# Shielding Federated Learning: Aligned Dual Gradient Pruning Against Gradient Leakage

## Abstract

Federated learning (FL) is a distributed learning framework that claims to protect user privacy. However, gradient inversion attacks (GIAs) reveal severe privacy threats to FL, which can recover the users' training data from outsourced gradients. Existing defense methods adopt different techniques, e.g., differential privacy, cryptography, and gradient perturbation, to against the GIAs. Nevertheless, all current state-of-the-art defense methods suffer from a trade-off between privacy, utility, and efficiency in FL. To address the weaknesses of existing solutions, we propose a novel defense method, Aligned Dual Gradient Pruning (ADGP), based on gradient sparsification, which can improve communication efficiency while preserving the utility and privacy of the federated training. Specifically, ADGP slightly changes gradient sparsification with a stronger privacy guarantee. Through primary gradient parameter selection strategies during training, ADGP can also significantly improve communication efficiency with a theoretical analysis of its convergence and generalization. Our extensive experiments show that ADGP can effectively defend against the most powerful GIAs and significantly reduce the communication overhead without sacrificing the model's utility.

## 1 Introduction

Federated learning (FL) [1] is a distributed learning framework, where multiple users train and send their gradients of the local models to the server without sharing their local data [1, 2, 3]. FL claims to protect the users' training data since the users do not need to share local data with the server directly. However, recent studies reveal that gradients can be used to recover the original training data information via gradient inversion attacks (GIAs) [4, 5]. To against GIAs, a large number of studies have been proposed, where they leverage the advanced privacy protection techniques, such as differential privacy (DP) [6], cryptography [7, 8, 9] and gradient perturbation [10, 11, 12]. However, none of the existing defense methods could take care of all privacy, utility, and efficiency difficulties in FL. For example, DP and cryptography-based methods could effectively defend GIAs, but sacrifice either the utility or efficiency respectively [6, 7, 8, 9]. In order to achieve better utility and efficiency in FL, perturbation-based methods design various gradient perturbations [10, 11, 12], but all existing perturbation-based methods could only defend one or two kinds of GIAs in practice.

Fox example, recent perturbation-based defense methods (*i.e.*, Precode [12], Soteria [10], and ATS [11]) can effectively defend against optimization-based GIAs [5, 13, 14, 15], but fail to work against the active GIAs [16, 17, 18]. On the contrary, the classic Top-$k$ based gradient sparisification method [19, 20] is generally considered as a bad privacy protection solution on optimization-based GIAs, but in fact performs much better than recent defense methods under the active attack from our experiments as shown in Table 2. The new findings inspire us to seek for a more practical perturbation-based defense against both optimization-based and active GIAs.

Submitted to 36th Conference on Neural Information Processing Systems (NeurIPS 2022). Do not distribute.

Table 1: Comparison of our method with existing privacy-preserving FL methods. Note: ✓ represents the scheme has a high guarantee for the property, while ✗ represents otherwise.

| Defense | Privacy | | | | | Utility | Efficiency |
|---|---|---|---|---|---|---|---|
| | Analytical attack | Optimization attack | | Active server attack | | | |
| | R-GAP [22] | DLG [4] | IVG [5] | Curious [16] | Rob [21] | | |
| Precode [12] | ✓ | ✓ | ✓ | ✗ | ✗ | ✗ | ✗ |
| ATS [11] | ✗ | ✓ | ✗ | ✗ | ✗ | ✓ | ✗ |
| Soteria [10] | ✓ | ✓ | ✓ | ✗ | ✗ | ✓ | ✗ |
| DP [23] | ✓ | ✓ | ✓ | ✓ | ✓ | ✗ | ✗ |
| Top-$k$ [19] | ✓ | ✓ | ✗ | ✓ | ✓ | ✓ | ✓ |
| ADGP (Ours) | ✓ | ✓ | ✓ | ✓ | ✓ | ✓ | ✓ |

In this paper, we propose a new gradient pruning based method, **A**ligned **D**ual **G**radient **P**runing (ADGP). Specifically, ADGP consists of two components: dual gradient pruning (DGP) and gradient location bounding. Dual gradient pruning is a novel gradient sparsification technique, which removes top-$k_1$ largest gradient parameters and the bottom-$k_2$ smallest gradient parameters from the local model. DGP leads to a strong privacy protection against both optimization-based and active GIAs. To further reduce the expensive download costs caused by the asymmetric gradient selection among different users, we propose gradient location bounding strategy to make the aggregated global gradient stay in the same sparsified region. In summary, ADGP achieves a better utility and privacy trade-off, increases FL system efficiency, and is robust against active attacks.

Furthermore, we give the theoretical analysis of ADGP, which proves the reconstruction error is proportional to gradient distance. So removing larger gradient parameters could enlarge the gradient distance resulting in a low reconstruction error. However, removing larger gradients will significantly impact the model's utility. Thus, to improve the sparsification ratio, which is essential to robustness against active attack [21, 16], we also remove the model parameters with smaller gradients. In such a way, our method could significantly mitigate GIAs without affecting the model's utility.

We conduct extensive experiments over multiple datasets and models to evaluate our method. The quantitative and visualized results show that our design can effectively make recovered images recognizable under different attacks, and reduce nearly half of the communication costs. Our contributions are as follows: 1) We revisit gradient sparsification to show its potential on mitigating GIA; 2) We propose an improved gradient pruning strategy to provide sufficient privacy guarantee while balancing the model accuracy and the system efficiency; 3) We conduct extensive experiments to show that our design outperforms perturbation-based defense methods *w.r.t* privacy protection, model accuracy, and system efficiency.

## 2  Related work

Federated learning [1, 3] is considered to be a privacy-preserving framework for distributed machine learning as the training data is not directly outsourced. However, the emerging of GIAs [4, 5, 21, 16, 22, 24, 25, 26, 27] shatters this conception. It has been proved that the attacker (*e.g.*, a curious server) can easily recover the private training data from gradient information to a great extent. The privacy guarantee of federated learning urgently needs to be strengthened.

**Cryptographic Defense.**   Traditionally, there are two approaches to construct privacy-preserving federated learning: using DP to disturb gradients [6, 23, 28, 29, 30] or using cryptographic tools to perform secure aggregation [7, 8, 9, 31, 32]. DP [6] is a popular and effective privacy protection mechanism by adding random noise to the raw data, but it is well known that the noises introduced by DP can greatly degrade the model accuracy when meaningful privacy is enforced [33]. Cryptographic-based secure aggregation can guarantee both privacy and accuracy simultaneously, but incurs expensive computation and communication costs [34]. Using the shuffle model [35, 36] can only provide anonymity. Moreover, it totally changes the system model of FL since an additional semi-trusted third party is introduced to work cooperatively with the server.

**Gradient Perturbation Defense.**   Recently, researchers have begun to explore the possibility of constructing new gradient perturbation mechanisms to better balancing privacy and accuracy. Sun *et al.* [10] proposed Soteria, a scheme that perturbs the representation of training data by pruning the

gradients of a single fully connected layer. Gao *et al.* [11] proposed ATS, a training data augmentation policy by transforming original sensitive images into alternative inputs, to reduce the visibility of reconstructed images. Scheliga *et al.* [12] presented Precode to extend the model architecture by using variational bottleneck (VB) [37] to prevent attackers from obtaining optimal solutions to reconstructed data. These defenses work well against GIAs in the semi-honest setting [4, 5, 38, 13], but fail to protect privacy when an active server modifies the model to launch GIAs [21, 16]. Moreover, these works suffer from high computation costs or huge communication burden.

**Gradient Sparsification Defense.** From an independent research domain, gradient sparsification has been commonly used for saving communication bandwidth. The most common sparsification strategy is Top-$k$ selection, which selects top $k$ gradient parameters with the largest absolute values [19, 20]. It has been widely proved that gradient sparsification provides very limited privacy protection ability [4, 10, 11, 12, 39] unless a high pruning ratio (*e.g.*, removing 99% of the gradients) is used at the cost of 10% accuracy drop [39]. However, we emphasize that this is misunderstood as they only consider the Top-$k$ sparsification strategy that has never received an in-depth investigation in the field of security. It is originally designed for improving system efficiency, thus a direct application inherently suffers from many weaknesses. As shown in Section 4, a slight modification can unleash the potential of gradient sparsification to provide a strong privacy guarantee.

# 3 Threat Model and Attacks

In this work, we consider a strong threat scenario, where an active server, after receiving gradients from users, tries to reconstruct the local training data and is motivated to modify model parameters in each iteration to strengthen the attack performance. Note that the server also wants to obtain a high-quality global model with high accuracy. More specifically, we consider the following three kinds of GIAs:

**Analytical attack.** Analytical attack exploits the structure of the gradients to recover the input, such as using gradient bias terms [40]. Recently proposed R-GAP attack [22] exploits the recursive relationship between gradient layers to solve the input. An effective analytical attack depends on the specific structure and parameters of gradients.

**Optimization attack.** Optimization attack is first proposed using L-BFGS optimizer to solve $\min ||\frac{\partial l(\mathbf{x},\mathbf{y})}{\partial \mathbf{W}} - \frac{\partial l(\mathbf{x}^*,\mathbf{y}^*)}{\partial \mathbf{W}}||_2^2$ and gets dummy data $\mathbf{x}^*$ and dummy label $\mathbf{y}^*$, where $\mathbf{y}$ is the label of $\mathbf{x}$ [4]. The state-of-the-art optimization attack method IVG [5] uses Adam to optimize the cosine distance and has been widely used to evaluate defense works [10, 11, 12].

Despite different optimizers can be used to achieve better attack quality [5, 13, 14, 15], the existing attacks are all measured by the distance between the generated gradients $\nabla \mathbf{W}^*$ and the original gradients $\nabla \mathbf{W}$. We therefore propose a general definition for optimization attack to better evaluate its performance. As shown in Definition 1, a smaller $\varepsilon$ indicates a stronger optimization attack.

**Definition 1.** *An optimization attack is a $(\varepsilon, \delta)$-attack, if it satisfies:*

$$\mathbb{P}(\mathbb{E}(\mathcal{D}_{(\nabla \mathbf{W}, \nabla \mathbf{W}^*)}) \leq \varepsilon) \geq 1 - \delta. \tag{1}$$

*where $\mathbb{P}$ represents the probability, $\mathbb{E}$ represents the expectation, $\mathcal{D}$ is the distance function commonly instantiated with Euclidean or cosine distance.*

**Active server attack.** In this kind of attack, the server can actively modify the global model to realize a better attack result rather than honestly executing the protocols [16, 17, 18]. Recently proposed Rob attack [21] adds imprint modules to the model and uses the difference between the gradient parameters in adjacent rows of the imprint module to recover the data, achieving the best attack effect in the literature.

# 4 Aligned Dual Gradient Pruning

## 4.1 Analysis of Gradient Sparsification

We owe the failure of common Top-$k$ gradient sparsification methods to two reasons: 1) the distance between the Top-$k$ sparsified (*i.e.*, perturbed) gradient $\mathbf{g}$ and the real gradient $\nabla \mathbf{W}$ is small; and 2) large gradient parameters in $\nabla \mathbf{W}$ also reveal label information about user data.

To explain the first reason, we investigate the relationship between the reconstruction error of user data and distance of perturbed gradient v.s. real gradient, as shown in Proposition 1.

**Proposition 1.** *For any given input* $\mathbf{x}$ *and shared model* $\mathbf{W}$*, the distance between the recovered data* $\mathbf{x}'$ *and the real data* $\mathbf{x}$ *is bounded by:*

$$||\mathbf{x} - \mathbf{x}'||_2 \geq \frac{||\nabla\mathbf{W} - \mathbf{g}||_2}{||\partial\varphi(\mathbf{x}, \mathbf{W})/\partial\mathbf{x}||_2}, \tag{2}$$

*where* $\varphi$ *is the mapping from* $\mathbf{x}$ *to* $\nabla\mathbf{W}$*, i.e., the reconstruction quality is limited by* $||\nabla\mathbf{W} - \mathbf{g}||_2$.

The proof of the above proposition is moved to the supplementary due to space limit (the same hereinafter). From this Proposition, it is clear that the reconstruction error is proportional to the gradient distance $||\nabla\mathbf{W} - \mathbf{g}||_2$, i.e., effective defense methods should enlarge the gradient distance as much as possible. However, for the Top-$k$ based gradient sparsification [19, 20], the $k$ largest parameters are retained, making the gradient distance small by nature.

To explain the second reason, we consider a $L$-layer perceptron model trained with cross-entropy loss for classification. Let a column vector $\mathbf{r} = [r_1, r_2, \ldots, r_n]$ be the logits (the output of the $L$-th linear layer) that input to the softmax layer, the confidence score probability vector is thus $\left[\frac{e^{r_1}}{\sum_j e^{r_j}}, \frac{e^{r_2}}{\sum_j e^{r_j}}, \cdots, \frac{e^{r_n}}{\sum_j e^{r_j}}\right]$ and the succinct form of the cross-entropy loss becomes $\ell(\mathbf{x}, y) = -\log(\frac{e^{r_y}}{\sum_j e^{r_j}})$. Focus on the $L$-th layer $\mathbf{W}^L\mathbf{x} + \mathbf{b}^L = \mathbf{r}$, it is easy to find

$$\frac{\partial\ell(\mathbf{x}, y)}{\partial b_i} = \frac{\partial\ell(\mathbf{x}, y)}{\partial r_i} \cdot \frac{\partial r_i}{\partial b_i} = \frac{\partial\ell(\mathbf{x}, y)}{\partial r_i} = \frac{e^{r_i}}{\sum_j e^{r_j}} - \mathbb{I}_{i=y}, \tag{3}$$

and

$$\nabla\mathbf{W}^L = \frac{\partial\ell(\mathbf{x}, y)}{\partial r} \cdot \mathbf{x}^T = [\frac{\partial\ell(\mathbf{x}, y)}{\partial r_1}, \cdots, \frac{\partial\ell(\mathbf{x}, y)}{\partial r_n}] \cdot \mathbf{x}^T. \tag{4}$$

For a given $\mathbf{x}$ (and so $\mathbf{x}^T$ is fixed), the magnitude of certain elements of the gradient matrix $\nabla\mathbf{W}^L$ (i.e., the $i$-th row) is particularly large if $i$ is the true label of the training data $\mathbf{x}$ due to reason that $|\frac{\partial\ell(\mathbf{x}, y)}{\partial r_i}| = \sum_{j\neq i}|\frac{\partial\ell(\mathbf{x}, y)}{\partial r_j}|$.

To summarize, due to the above two reasons, we conclude that common Top-$k$ gradient sparsification cannot provide sufficient protection for user data against passive optimization attacks. From another point of view, a sufficient gradient sparsification ratio also plays an important role in defending against active server attacks. As mentioned in Section 3, active attackers can exploit the correspondence of partial gradient parameters to recover the real data. So, the gradient sparsity rate will directly destroy the relationship among gradient parameters constructed by the active attacker. Intuitively, the higher the sparsity rate, the more severe the impact. As will be validated in Section 6, a higher sparsity rate can prevent the attacker from obtaining useful gradient information.

## 4.2 Dual Gradient Pruning

Generally speaking, large gradient parameters of local model need to be removed to make the gradient difference larger, but the difference should also be appropriately bounded to maintain high model accuracy. Moreover, it is also necessary to delete small gradient parameters to achieve a high sparsification ratio. With these observations, we propose dual gradients pruning (DGP), a new parameter selection strategy for gradient sparsification.

The users first sort the absolute values of all Size($\nabla\mathbf{W}$) local gradient parameters in the descending order. Let $\mathcal{T}_{k_1}(\nabla\mathbf{W})$ represent the set of top-$k_1$ elements of $\nabla\mathbf{W}$, $\mathcal{B}_{k_2}(\nabla\mathbf{W})$ represent the set of its bottom-$k_2$ elements. Then the users remove $\mathcal{T}_{k_1}(\nabla\mathbf{W})$ and $\mathcal{B}_{k_2}(\nabla\mathbf{W})$ from $\nabla\mathbf{W}$ for gradient sparsification. Note that we set $p = k_2/k_1$ as a hyperparameter to regulate the trade-off between privacy and accuracy. Clearly, even with a fixed value $p$, different user will have different sets of $\mathcal{T}_{k_1}(\cdot)$ and $\mathcal{B}_{k_2}(\cdot)$ because their respective local models could be different from each other.

We emphasize that although such dual gradients pruning strategy is very simple, it can significantly mitigate GIAs without affecting the model accuracy. A rigorous security proof is shown in Section 5, and experimental results can be found in Section 6.

---

**Algorithm 1:** Aligned Dual Gradient Pruning (ADGP)

**Input** : Original gradient matrix $\nabla \mathbf{W}$, location binary matrix $\mathcal{I}$, values of $k_1$ and $k$
**Output** : Sparsified gradient matrix $\mathbf{g} = \{\mathbf{g}^1; \cdots; \mathbf{g}^L\}$
**for** $l \leftarrow 1$ *to* $L$ **do**
$\quad$ Remove parameters in $\mathcal{T}_{k_1}(\nabla \mathbf{W}^l)$ from $\nabla \mathbf{W}^l$
$\quad$ Keep parameters in $\nabla \mathbf{W}^l$ when location is in $\mathcal{I}$, and discard all other parameters
$\quad$ Upload $\mathbf{g}^l = \mathcal{T}_k(\nabla \mathbf{W}^l)$ to the server

---

**Algorithm 2:** A Complete Illustration of our Defense

**Input** : Initial global model $\mathbf{W}^0$, value $k$ and $k_1$, total rounds $T$, total users $N$
**Output** : Shared global model $\mathbf{W}^T$
Set $\mathbf{e}^0 = 0$
**for** $t \leftarrow 0$ *to* $T - 1$ **do**
$\quad$ Randomly select a user to broadcast the location matrix $\mathcal{I}^t$ of its parameter set $\mathcal{T}_{2k}$
$\quad$ **for** $i \leftarrow 1$ *to* $N$ **do**
$\quad\quad \mathbf{P}_i^t = \nabla \mathbf{W}_i^t + \mathbf{e}_i^t$
$\quad\quad \mathbf{g}_i^t = \text{ADGP}(k_1, k, \mathbf{P}_i^t, \mathcal{I}^t)$
$\quad\quad \mathbf{e}_i^{t+1} = \mathbf{P}_i^t - \mathbf{g}_i^t$
$\quad$ Sever side aggregation:
$\quad \mathbf{W}^{t+1} = \mathbf{W}^t - \eta \frac{\sum_{i=1}^{N} \mathbf{g}_i^t}{N}$

---

## 4.3 A Complete Illustration of Our Method

Although dual gradient pruning provides a sufficient privacy guarantee as well as reduces upload cost of users, users' download costs could still be expensive. This is because different users have different sets of $\mathcal{T}_{k_1}(\cdot)$ and $\mathcal{B}_{k_2}(\cdot)$ when sparsifying their own local gradients, which will ultimately make the global gradient become dense after aggregation.

We thus propose aligned dual gradient pruning (ADGP), an improved scheme to align the selected gradients across different users. Similar to DGP, for best privacy, each user will still firstly identify his top-$k_1$ gradients location set $\mathcal{T}_{k_1}$. Different from DGP, ADGP also wants to save users' download cost by ensuring that all users' uploaded sparsified gradients reside in the same location set. This is achieved by randomly selecting a user, who identifies a top-$2k$ ($k_1 < k$) location set $\mathcal{T}_{2k}$ (represented with a binary matrix $\mathcal{I}$) and broadcasts $\mathcal{I}$ to all other users. Note that $\mathcal{T}_{k_1} \subset \mathcal{T}_{2k}$ is not necessary true. Upon receiving $\mathcal{I}$, each user first discards gradient parameters in $\mathcal{T}_{k_1}$ and then only transmits the $k$ largest gradient parameters whose locations belong to $\mathcal{I}$. After aggregation at the server side, users only need to download global gradients' parameters associated with $\mathcal{I}$. A detailed illustration of ADGP is shown in Algorithm 1.

For ADGP pruning, in each FL iteration round, all gradient parameters whose locations are outside of $\mathcal{I}$ will not participate the current round global model aggregation. In the extreme case, $\mathcal{I}$ can remain static for all iteration rounds and the local accumulated error (accumulated unused local gradient parameters) becomes large, thus hindering global model convergence. To reduce this negative impact and increase convergence speed, we design an error feedback mechanism. In particular, at the iteration round $t$, after user $i$ obtaining his local gradient $\nabla \mathbf{W}_i^t$, he will combine $\nabla \mathbf{W}_i^t$ with an error term accumulated in the previous $(t - 1)$ rounds before performing the ADGP sparsification pruning. A complete illustration of our method is shown in Algorithm 2.

## 5 Theoretical Analysis

This section presents the security analysis with regard to passive attacks (i.e., analytical and optimization attacks presented in Section 3), as well as the generalization and convergence analyses of the proposed ADGP algorithm. Following the literature studies in [41, 42], for a given $L$-layer centralized model, we model the first $(L - 1)$ layers as a robust feature extractor of any input sample.

Thus, the function of this model is characterized by $f(x|\mathbf{W}) = \mathbf{W}x + \mathbf{b}$, and the optimization objective is the loss $\ell(\mathbf{x}, y)$ (such as cross-entropy or L2 loss). To facilitate analyses and following literature studies [19, 41, 43, 44], the assumptions about the smoothness of DGP, ADGP and $l$, as well as the variance of the stochastic gradient are employed.

**Assumption 1.** *The pruning mechanisms* $\mathrm{DGP}(k_1, k_2, \nabla\mathbf{W}^t)$ *and* $\mathrm{ADGP}(k_1, k, \nabla\mathbf{W}^t, \mathcal{I}^t)$ *are both bi-Lipschitz, so the following conditions hold:*

$$||\nabla\mathbf{W} - \mathrm{DGP}(k_1, k_2, \nabla\mathbf{W})||_2^2 = ||\mathrm{DGP}(0, 0, \nabla\mathbf{W}) - \mathrm{DGP}(k_1, k_2, \nabla\mathbf{W})||_2^2 \geq \gamma_1 ||\nabla\mathbf{W}||_2^2,$$
$$||\nabla\mathbf{W} - \mathrm{ADGP}(k_1, k, \nabla\mathbf{W}^t, \mathcal{I}^t)||_2^2 \leq \gamma_2 ||\nabla\mathbf{W}||_2^2,$$

*where* $\gamma_1$ *is a constant determined by* $k_1$ *and* $k_2$, *and* $\gamma_2$ *is a constant determined by* $k_1$ *and* $k$.

**Assumption 2.** *The objective function* $l : R^d \to R$ *has a low bound* $l^*$ *and it is Lipschitz-smooth, i.e., for any* $x_1, x_2$, $||\nabla l(x_1) - \nabla l(x_2)||_2 \leq K ||x_1 - x_2||_2$ *and* $l(x_1) \leq l(x_2) + \langle \nabla l(x_2), x_1 - x_2 \rangle + \frac{K}{2}||x_1 - x_2||_2^2$.

**Assumption 3.** *The full gradient* $\nabla l(\mathbf{W}^t)$ *is bounded, i.e.,* $||\nabla l(\mathbf{W}^t)||_2^2 \leq G^2$, *and the federated stochastic gradient* $\nabla\mathbf{W}_i^t$ $(i = [1, N])$ *is the unbiased estimation of the full gradient* $\nabla l(\mathbf{W}^t)$, *i.e.,* $\mathbb{E}(\nabla\mathbf{W}_i^t) = \nabla l(\mathbf{W}^t)$. *Moreover, the variance between* $\nabla\mathbf{W}_i^t$ *and* $\nabla l(\mathbf{W}^t)$ *is bounded:* $\mathbb{E}||\nabla\mathbf{W}_i^t - \nabla l(\mathbf{W}^t)||_2^2 \leq \sigma^2$.

**Security Analysis.** It is noted that, for the same sparsification ratio, user's uploaded gradient parameters from ADGP is generally smaller than that from DGP. Indeed, the uploaded gradient parameters from both methods are the same only when $\mathcal{T}_{k_1} \subset \mathcal{T}_{2k}$ holds. From this observation and referring to Proposition 1, DGP is the security lower bound of our design for privacy protection. So, our focus is the security analysis of DGP. As shown in the theorem below, we prove that DGP achieves a stronger privacy protection in the sense of Definition 1.

**Theorem 1.** *For any* $(\varepsilon, \delta)$ *optimization attack, under the presence of* DGP, *it will be degenerated to* $(\varepsilon + \sqrt{\gamma_2}||\nabla\mathbf{W}||_2, \delta)$*-attack if* $\mathcal{D}$ *is measured by Euclidean distance, and degenerated to* $(1 - \sqrt{\gamma_1}(1 - \varepsilon), \delta)$*-attack if* $\mathcal{D}$ *is measured by cosine distance.*

The Theorem is based on Assumption 1 about DGP. It reveals that, with the same successful chance $1 - \delta$, DGP weakens the attacker's capability to optimize a better estimation of the true $\nabla\mathbf{W}$.

**Generalization and Convergence Analyses.** The generalization analysis aims to quantify how the trained model performs on the test data, and it is achieved by analyzing the how ADGP affects the properties of the optima reached (without gradient pruning) [41, 42]. Assisted with Assumption 3 and Assumption 1 about ADGP gradient pruning, the following Lemma can be obatined.

**Lemma 1.** *Let* $\mathbf{e}^t = \sum_{i=1}^{N} \mathbf{e}_i^t / N$ *be the averaged accumulated error among all users at iteration* $t$, *the expectation of the norm of* $\mathbf{e}^t$ *is bounded, i.e.,*

$$\mathbb{E}||\mathbf{e}^t||_2^2 \leq \frac{\gamma_2}{2}\left(\frac{2 + \gamma_2}{1 - \gamma_2}\right)^2 (G^2 + \sigma^2). \tag{5}$$

Note that the difference between the averaged pruned gradient $\mathbf{g}^t = \sum_{i=1}^{N} \mathbf{g}_i^t / N$ and the averaged Fed-SGD gradient $\nabla\mathbf{W}^t = \sum_{i=1}^{N} \nabla\mathbf{W}_i^t / N$ is simply $||\sum_{i=0}^{T-1}(\nabla\mathbf{W}^t - \mathbf{g}^t)||_2^2 = ||\mathbf{e}^T||_2^2$. So the lemma above indicates that the accumulated gradient difference between our algorithm and Fed-SGD is bounded. That said, the optima reached by ADGP and the optima reached by Fed-SGD will eventually be the same if the algorithm converge. Armed with Lemma 1 and based on Assumptions 1, 2 and 3, we demonstrate the convergence of the our algorithm.

**Theorem 2.** *The averaged norm of the full gradient* $\nabla l(\mathbf{W}^t)$ *derived from centralized training is correlated with the our algorithm as follows:*

$$\frac{\sum_{t=0}^{T-1} \mathbb{E}||\nabla l(\mathbf{W}^t)||_2^2}{T} \leq 4\frac{l^0 - l^*}{\eta T} + 4\eta^2 K^2 \frac{\gamma_2}{2}\left(\frac{2 + \gamma_2}{1 - \gamma_2}\right)^2 (G^2 + \sigma^2) + 2K\eta(G^2 + \frac{\sigma^2}{N}), \tag{6}$$

*where* $l^0$ *is the initialization of the objective* $l$, *and* $\eta$ *is the learning rate.*

The immediate implication of Theorem 2 is that, with an appropriate learning rate $\eta$, ADGP converges similar to Fed-SGD (slower by a negligible term $\mathcal{O}(\frac{1}{T})$), as shown in Corollary 1.

Table 2: Evaluation of defense performance under three attacks.

| Attack | Metric | Baseline | Precode | DP | Soteria | ATS-I | ATS-II | Top-$k$ | Ours |
|---|---|---|---|---|---|---|---|---|---|
| **IVG** | PSNR ($\downarrow$) | 34.8805 | 9.6441 | **6.9554** | 9.2447 | 16.6894 | 31.3200 | 14.1338 | 7.6192 |
| | LPIPS($\uparrow$) | 0.0016 | 0.4473 | **0.5504** | 0.3774 | 0.1621 | 0.0015 | 0.2754 | 0.4829 |
| | SSIM ($\downarrow$) | 0.9273 | 0.4793 | **0.2451** | 0.4173 | 0.6851 | 0.9189 | 0.5336 | 0.2923 |
| **R-GAP** | PSNR ($\downarrow$) | 36.7656 | - | **5.0691** | 5.1817 | 10.8442 | 42.0900 | 5.1017 | 5.1196 |
| | LPIPS($\uparrow$) | 0.0007 | - | 0.3621 | 0.3532 | 0.2094 | 1.8e-05 | 0.4817 | **0.4863** |
| | SSIM ($\downarrow$) | 0.9307 | - | 0.2483 | 0.2124 | 0.3962 | 0.9121 | 0.2027 | 0.1928 |
| **Rob** | PSNR ($\downarrow$) | 102.8838 | 109.6553 | **8.7491** | 102.8838 | 9.6166 | 115.9886 | 13.0685 | 13.0804 |
| | LPIPS($\uparrow$) | 0.0960 | 0.1488 | **1.3434** | 0.0960 | 0.6410 | 0.0486 | 0.8920 | 0.9184 |
| | SSIM ($\downarrow$) | 0.8969 | 0.8440 | 0.2064 | 0.8969 | 0.2545 | 0.9490 | 0.0428 | **0.0229** |
| Final model accuracy | | **93.4400** | 93.1699 | 86.8900 | 93.2300 | 93.3900 | 93.3900 | 93.2099 | 93.1700 |

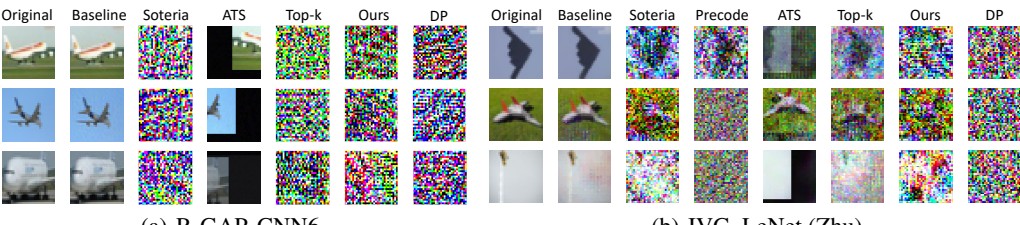

(a) R-GAP, CNN6        (b) IVG, LeNet (Zhu)

Figure 1: Visualization of the reconstructed data under R-GAP and IVG attacks.

**Corollary 1.** *Let* $\eta = \sqrt{\frac{l^0 - l^*}{KT(G^2 + \sigma^2/N)}}$, *we have*

$$\frac{\sum_{t=0}^{T-1} \mathbb{E}||\nabla l(\mathbf{W}^t)||_2^2}{T} \leq 6\sqrt{\frac{l^0 - l^*}{KT(G^2 + \sigma^2/N)}} + \mathcal{O}(\frac{1}{T}). \tag{7}$$

# 6 Experiments: Privacy-Accuracy-Efficiency Tradeoff

## 6.1 Experimental Setup

**Datasets and models.** We conduct experiments on CIFAR10 with LeNet (Zhu) [5], CIFAR10 [45] with CNN6 and CIFAR100 [45] with LeNet (Zhu) and ResNet18 respectively. We run these experiments in a pytorch environment by using a single RTX 2080 Ti GPU and 2.10GHz CPU.

**Evaluation Metrics.** We quantify the privacy effect of defenses, follow [39, 46], we visualize the reconstructed data and use learned perceptual image patch similarity (LPIPS), peak signal-to-noise ratio (PSNR), structural similarity (SSIM) to measure the quality of the recovered data. A better defense scheme should has a larger LPIPS ($\uparrow$), smaller peak signal-to-noise ratio (PSNR) ($\downarrow$) and structural similarity (SSIM) ($\downarrow$).

**Attack methods.** We evaluate our defense against IVG attack [5], R-GAP attack [22], and Rob attack [21], which represent three kinds of state-of-the-art GIAs, as illustrated in Section 3. All these attacks are implemented strictly following the original settings, *i.e.*, IVG is evaluated on CIFAR10 with LeNet (Zhu), R-GAP is evaluated on CIFAR10 with CNN6, Rob attack is evaluated on CIFAR100 with LeNet (Zhu). More settings for attacks are shown in the supplementary.

**Defense methods.** We compare our method with five state-of-the-art defenses: Soteria [10], ATS [11], Precode [12], Differential Privacy (DP) [2], and Top-$k$ based gradient sparsification[1] [19]. Besides, we set Fed-SGD [3] as the baseline that adopts no defenses. Following the DP design in [2], we use the Gaussian differential privacy mechanism with $\varepsilon = 10.7, \delta = 10^{-5}$, which is the suggested best setting for the privacy-accuracy trade-off and can make most models converge. When quantifying the defense performance of ATS, we not only evaluate the similarity between the raw images and the recovered data (ATS-I), but also evaluate the similarity between the disturbed training

---

[1]Hereinafter, we abuse the notion of $k$ to denote the send rate $(k/\text{Size}(\nabla\mathbf{W})) \times 100\%$ since it will not cause ambiguity. And the sparse ratio is 1-k. The smaller the ratio $k$ is, the better communication efficiency.

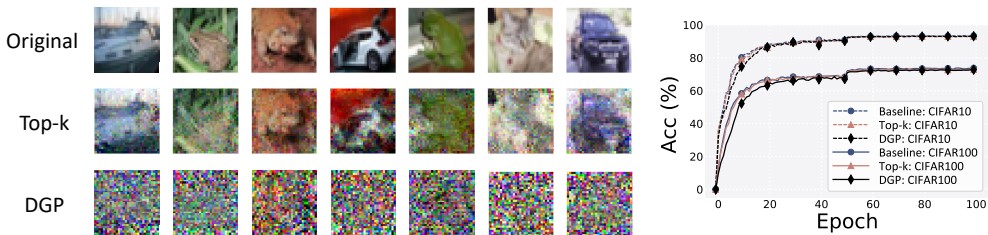

(a) Recovered images under IVG attack with ResNet18 on CIFAR100     (b) ResNet18 on CIFAR dataset

Figure 2: A detailed comparison between Top-$k$ and our DGP on privacy and model accuracy.

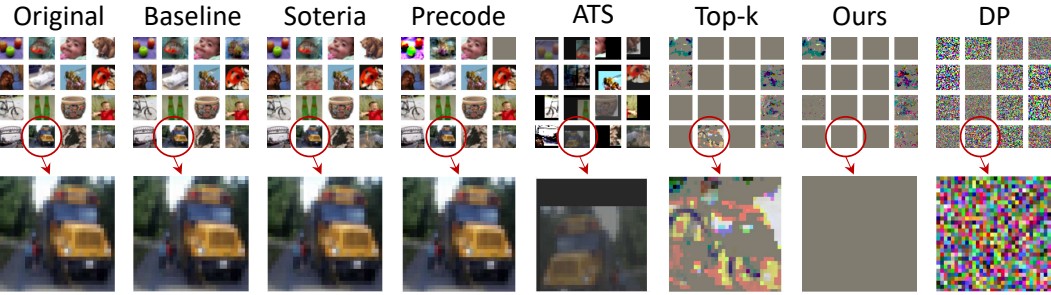

Figure 3: Visualization of reconstructed images under Rob attack with batchsize=16.

images (*i.e.*, the real inputs) and the recovered data (ATS-II). We set the send rate $k = 0.2$ and the regulation hyperparameter $p = 15$. The supplementary gives more experiments under different $p$ and $k$. The rest defense schemes remain the original settings.

## 6.2   Defense Performance Evaluation

Table 2 shows the defense performance with PSNR, SSIM, and LPIPS under three attacks. The results show that ATS, Soteria, Precode perform poorly under Rob attack, while Top-$k$ is vulnerable to IVG attack although it performs better under Rob attack. In most cases, our scheme performs comparably with DP and outperforms all the other defenses. More evaluation results under Rob attack are presented in our supplementary.

We also visualize the reconstructed images in order to perceptually demonstrate the defense performance. Figure 1(a) shows the the recovered images against R-GAP and IVG attacks. We can see that all the existing defenses can well defend against R-GAP attack except ATS because it does not damage the gradient structure, proving that a slight perturbation on gradients can mitigate the analytical attacks easily. We are not able to provide the result of Precode because its VB operation destroys the model structure thus analytical attack R-GAP cannot be implemented. In Figure 1(b), recovered images under IVG attack are presented. We can find that the attacker can still recover the outline of inputs with ATS and Top-$k$ defenses. DP, Soteria, Precode, and our scheme can still make the recovered images unrecognizable. Figure 3 evaluates the defenses against Rob attack. It shows that ATS, Soteria, and Precode fail to work and most inputs can be reconstructed.

In Rob attack, the attacker uses the gradient of the imprint module to reconstruct the training data. Our method, Top-$k$, and DP can effectively defend against Rob attack because the gradients of all layers are sparsed or perturbed, including those of the malicious imprint modules. However, we emphasize that the main weakness of Top-$k$ is its vulnerability to optimization attacks (*e.g.*, IVG), as widely demonstrated in the literature [4, 10, 11, 39, 12]. We thus further evaluate Top-$k$ and our scheme under IVG attack with ResNet18 on CIFAR datasets. We set $k_1/\text{Size}(\nabla \mathbf{W}) = 0.05, k_2/\text{Size}(\nabla \mathbf{W}) = 0.75$.

## 6.3   Model Accuracy Evaluation

To evaluate model performance, we train ResNet18, LeNet (Zhu), VGG13_bn [47] on CIFAR10, CIFAR100 with ten users, respectively. We set epoch=100, the learning rate $\eta$=0.1 if epoch $\leq$ 50,

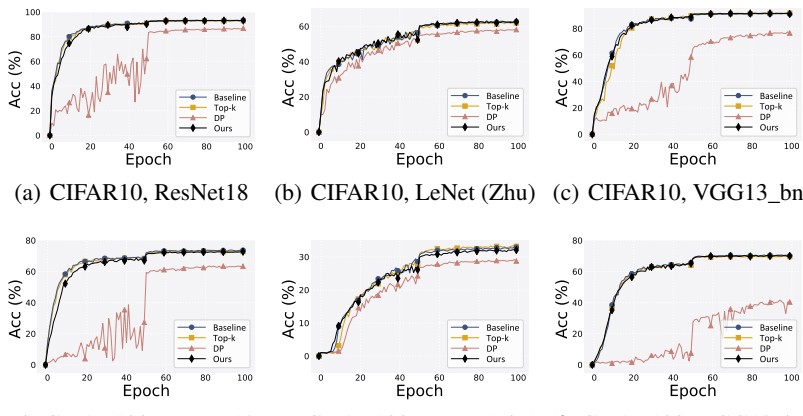

(a) CIFAR10, ResNet18   (b) CIFAR10, LeNet (Zhu)   (c) CIFAR10, VGG13_bn

(d) CIFAR100, ResNet18   (e) CIFAR100, LeNet (Zhu)   (f) CIFAR100, VGG13_bn

Figure 4: Evaluation of model accuracy with different datasets and model architectures.

Table 3: Commu. cost in one iteration (MB)

| Method | Baseline | Soteria | Precode | ATS | DP | Top-k | Ours |
|---|---|---|---|---|---|---|---|
| Resnet18 | 85.2506 | 85.2268 | 88.1644 | 85.2506 | 85.2506 | 43.7979 | 27.3067 |
| VGG13 | 71.8385 | 71.7318 | 74.8433 | 71.8385 | 71.8385 | 34.9625 | 22.9697 |
| LeNet | 0.1207 | 0.0764 | 1.8624 | 0.1207 | 0.1207 | 0.0493 | 0.0424 |

$\eta$=0.01 if epoch >50, and $\eta$=0.05 if epoch >70. We show in Table 2 the accuracy of ResNet18 over CIFAR10 under different defenses, and here we only compare our scheme with the baseline Fed-SGD, DP, and Top-$k$ since they perform best for privacy protection. Because [41] showed that the error feedback is beneficial to improve the model accuracy, even without using gradient sparsification. To give a fair comparison, we set the error feedback mechanism as the basic setting for all the defenses. The experimental results in Figure 4 show that we achieve similar model performance with the baseline, while DP, as expected, significantly damage the model accuracy.

## 6.4 Efficiency Evaluation

To clearly demonstrate the system efficiency, we evaluate the communication cost, which is obtained by computing the total overheads of sending updated gradients and receiving aggregated gradients. For ease of presentation, we only show the results for one iteration. As shown in Table 3, our scheme reduces more than half of the communication costs compared with existing defenses, and our gradient sparsification incurs negligible computation burden. The specific computation cost evaluation is presented in the supplementary.

## 7 Conclusions, Limitations, and Broader Impact

Our work firstly reveals the risks of privacy-preserving methods that only perturb the gradients of some layers. Through a comprehensive analysis of gradient inversion attacks, we show that it is necessary to perturb or sparse the gradients of each layer for privacy preservation. And considering the challenge of high communication cost in federated learning, we propose aligned dual gradient sparsification method to achieve the trade-off between privacy protection, model performance, and efficient communication, and give sufficient theoretical support. We hope that our newly proposed gradient sparisification method can shed new light on addressing privacy leakage concern as well as saving communication bandwidth.

In terms of limitations, the success of our scheme relies on selecting a reliable user to broadcast its gradient locations. Randomly selecting users may encounter malicious users that destroy the entire system. Our design is delegated to protecting privacy and has no negative societal impacts in practice.

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
