# OpenReview forum: "Shielding Federated Learning: Aligned Dual Gradient Pruning Against  Gradient Leakage"
_NeurIPS.cc/2022/Conference — NeurIPS 2022 Submitted_

### Official Review · Reviewer_U79T · 2022-07-09

**Rating:** 5
**Confidence:** 3
**Soundness:** 3 good
**Presentation:** 4 excellent
**Contribution:** 3 good

**Summary:**

This paper introduces two techniques, Dual Gradient Pruning (DGP) and Aligned Dual Gradient Pruning (AGDP) to protect against gradient inversion attacks. They describe DGP and present ADGP as an improved version of DGP that has the same security considerations but lower communication costs. They show that optimization attacks have a provable amount of degeneration under DGP and ADGP. Then they empirically evaluate their technique against a set of gradient inversion attacks and compare against other defenses like DP-SGD, Soteria and Precode. The authors show that their ADGP technique succeeds against all the attacks they try while still maintaining model quality and reducing communication costs.

**Questions:**

Does the analysis still hold if assumption 3 does not hold?
Did you try your technique on domains other than vision?



**Limitations:**

Negative impacts are adequately addressed.

**Strengths And Weaknesses:**

Strengths
- The authors present a very simple idea and run extensive experiments to show that their technique is superior to existing baselines
- DGP by itself seems like a new and somewhat unintuitive technique and the results from DGP would be a fairly strong contribution by itself.
- Empirical evaluation and comparison with other techniques is very thorough. The authors compare against most of the well-known baselines and try multiple attacks. The details of the empirical evaluation are also clearly specified in the main text.

Weaknesses
- Assumption 3 seems somewhat unrealistic. Assuming that each client in each round has data that provides an unbiased gradient means that the data has to be IID distributed across clients.
- ADGP is presented as saving communication overhead but this overhead is only saved if the same set of clients are used for every round of FL training. If the clients selected for round N are different from those selected for round N+1 then ADGP does not save any communication overhead. In production FL settings, clients are often selected from a large population, so the odds of the same client appearing in two back-to-back rounds are low.
- The authors only evaluate on vision models, but there are known gradient inversion attacks in NLP, speech etc. It would be interested to see if DGP works in those settings.

---

> ### Author Response · Authors · 2022-08-02
> **Response to Reviewer U79T**
>
> Thanks for your thoughtful comments. We provide the following responses for your concerns.
>
> >  W1: Assumption 3 seems somewhat unrealistic. Assuming that each client in each round has data that provides an unbiased gradient means that the data has to be IID distributed across clients.
>
> Re: In Assumption 3, we assume that stochastic gradients are the unbiased estimation of the full gradient. This assumption does not require the IID data distribution, and it can still be established in the non-IID scenario whenever federated learning is still valid. In particular, consider the basic Fed-SGD aggregation that all the clients' local gradient
>
>
> $$
> \Delta \mathbf{W}_i^t~(i=[1,N])
> $$
>  are averaged to obtain the aggregated gradient
> $$
> \nabla l(\mathbf{W}^t)
> $$
> at the central server. As
>
>
> $$
> t \rightarrow \infty
> $$
>  and the central model start to converge, it must be true that
>
>
> $$
> \lim_{t \rightarrow \infty} \left(\nabla \mathbf{W}_i^\infty - \nabla l(\mathbf{W}^\infty) \right)= 0
> $$
>  (otherwise, it is easy to deduce a contradiction that $\nabla l(\mathbf{W}^\infty)$ will not converge). In this sense, Assumption 3 is realistic for both IID and non-IID data distribution in the federated learning scenario.
> Note that our proposed methods DGP and ADGP are just variances of Fed-SGD, our converge analysis (Theorem 2) proves that our methods converge similar to Fed-SGD. Last-but-not-least, similar assumption used in this paper can be found in [1-2].
>
>
>
>
>
> > W2: ADGP is presented as saving communication overhead but this overhead is only saved if the same set of clients are used for every round of FL training. If the clients selected for round N are different from those selected for round N+1 then ADGP does not save any communication overhead. In production FL settings, clients are often selected from a large population, so the odds of the same client appearing in two back-to-back rounds are low.
>
> R2: There may be a misunderstanding here. The alignment strategy is used to save the download communication overhead. In each iteration, each selected client, although different from previous round, only needs to download partial of aggregated global model (determined by the location matrix), thus still enjoying the communication saving. Meanwhile, the saving of upload communication overhead is directly achieved since the uploaded gradients have been pruned in each round.
>
>
>
>
>  W3: The authors only evaluate on vision models, but there are known gradient inversion attacks in NLP, speech etc. It would be interested to see if DGP works in those settings.
>
>
>
> Re: We have not tried DGP on models other than vision yet. But it is interesting to extend our work to other domains, and we will show this in our revised version.
>
>
>
>   References:
>
> [1] Avdiukhin, Dmitrii, and Shiva Kasiviswanathan. "Federated learning under arbitrary communication patterns." International Conference on Machine Learning. PMLR, 2021.
>
> [2]Girgis, Antonious, et al. "Shuffled model of differential privacy in federated learning." International Conference on Artificial Intelligence and Statistics. PMLR, 2021.

---

### Official Review · Reviewer_CLEu · 2022-07-11

**Rating:** 4
**Confidence:** 4
**Soundness:** 2 fair
**Presentation:** 1 poor
**Contribution:** 1 poor

**Summary:**

This paper continues the research line of defending against data leakage in federated learning (FL). In particular, the paper argues against the conventional view that the top-k defenses do not work well, and proposed a simple modification, which is to also remove bottom gradients and then perform a calibration based on the accumulated error from the previous rounds to reduce communication cost. The paper compares the performance with five existing defenses on three existing attacks, and show that their method can achieve a better utility-privacy-communication tradeoff.

**Questions:**

1. The authors point out why top-k does not perform well: (1) top-k gradients are close to the real gradients and (2) large gradients leaks information of $y$. Both reasons only explain why top-k might be problematic, but how does removing bottom gradients address both problems? I do not find any insights that explain it, the only sentence I find that gives some justifications is "Moreover, it is also necessary to delete small gradient parameters to achieve a high sparsification ratio." (Section 4.2), which only says deleting small gradients can make sparsification ratio higher. But deleting any gradients can further sparsify, why small gradients? Please clarify.

2. Definition 1: Change $\mathcal{D}_{(\nabla W, \nabla W^*)}$ to $\mathcal{D}(\nabla W, \nabla W^*)$. Usually $D(.,.)$ represents the distance and there is no need of subscripting.

3. Proposition 1
- From the supplementary material, $\phi(x, W)$ is just $\nabla W$, you should define it explicitly in the main text. In addition, why write $\nabla W$ twice in the right-hand side of the inequality but in different forms (in both numerator and denominator)?
- Define $g = \phi(x', W)$ (taken from the supplementary material) explicitly. Propositions should be self-contained.
- The conclusion that "reconstruction error is proportional to the gradient distance $||\nabla W - g||_2$" seems questionable to me because the denominator is not a constant. In fact, if I understand it correctly, the denominator is the 2nd derivative of $W$ w.r.t $x$, i.e the Hessian norm, is it correct? If so, I do not see why the reconstruction error can be proportional to the numerator. Please let me know if I misunderstand it.

4. Algorithm 1: Where is the part that removes the bottom gradients $\mathcal{B}_{k_2}(\nabla W)$? It seems it only operates on the top gradients.

5. Assumption 1: $\gamma_1$ and $\gamma_2$ are constants determined by $k_1$, $k_2$, and $k$. What are their explicit definitions? Please write them out because they are used in main theorems and lemmas and should not be undefined.

6. Table 2
- The comparison of DP with other defenses are not apple-to-apple because the DP accuracy is much lower. I notice the reason you choose that DP budget was because it has been shown the best in the prior work. But in this particular evaluation, if you do not tune the DP budget to find a comparable accuracy, the privacy protection performance comparison is not fair.
- In R-GAP, the SSIM row, the best defense ("Ours") is not highlighted.

7. In general, the author should not make claims against the utility-privacy-communication tradeoff like "none of the existing defense methods could take care of all privacy, utility, and efficiency difficulties in FL" (taken from the introduction). Because there is a fundamental tradeoff and this paper (and maybe all papers) cannot remove the tradeoff. They can only make a better tradeoff but cannot make it go away. Therefore, I think it would be a more objective tone not to emphasize the tradeoff exists in the prior work (since it exists in any method) but rather the tradeoff is not good enough.

**Limitations:**

I do not have comments on the limitations.

**Strengths And Weaknesses:**

Strengths
- Important topic: the data leakage problem in FL is a practical concern
- The method is simple and easy to implement

Weaknesses
- The theoretical part has several places that are unclear and questionable to me (see Questions section)
- The evaluation section show some results that do not seem comparable (see Question section)

---

> ### Author Response · Authors · 2022-08-02
> **Response to Reviewer CLEu**
>
> Thanks for your thoughtful comments. We provide the following responses for your concerns.
>
>
>
>
>
> >  Q1: The authors point out why top-k does not perform well: (1) top-k gradients are close to the real gradients and (2) large gradients leaks information of . Both reasons only explain why top-k might be problematic, but how does removing bottom gradients address both problems?
>
>
>
> Re: There is a trade-off between privacy and accuracy. In order to defend against rob attacks, it is necessary to make a high gradient sparsity rate, \ie, pruning gradients as many as possible. However, removing too many large gradients, which usually contain important information, will cause a significant increase on the model accuracy. Therefore,  we choose to remove a certain number of small gradients to balance the privacy and accuracy.  Here we  briefly give some experimental results to illustrate this observation, as shown below.
>
>
>
> Evaluating ADGP under rob attack with different $k_1$ and $k_2$.
>
> | Imagenet     | LPIPS | SSIM   |
> | ------------ | ----- | ------ |
> | k1=10%,k2=0% | 0.47  | 0.632  |
> | k1=20%,k2=0% | 0.58  | 0.5361 |
> | k1=5%,k2=75% | 0.918 | 0.022  |
> | Cifar100     |       |        |
> | k1=10%,k2=0% | 0.12  | 0.6439 |
> | k1=20%,k2=0% | 0.24  | 0.4381 |
> | k1=5%,k2=75% | 0.482 | 0.2439 |
>
>
>
>
>
> > Q2, Q3.2, Q6.2 Q7:
>
> R2,6.1,7: Thanks for pointing these issues, we will revise all of them carefully.
>
>
>
> >  Q3.1: From the supplementary material, is just , you should define it explicitly in the main text. In addition, why write twice in the right-hand side of the inequality but in different forms (in both numerator and denominator)?
>
> Re:  We use  $\varphi(x,\mathbf{W})$ instead of $\nabla \mathbf{W}$ is to easily facilitate the proof of the relationship between $\nabla \mathbf{W}$ and $\mathbf{g}$. We will keep them consistent in the revised version.
>
>
>
> >  Q3.3: The conclusion that "reconstruction error is proportional to the gradient distance " seems questionable to me because the denominator is not a constant.
>
> Re: In Proposition 1, the denominator is actually a 2-norm of 2nd derivative of the Hessian matrix. For the given input x and the model W, the denominator is a constant. That is, we provide a specific privacy lower bound for a given x.
>
>
>
> >  Q4: Algorithm 1: Where is the part that removes the bottom gradients? It seems it only operates on the top gradients.
>
>
>
> R4: We send the top-k parameters  after removing the top-k1 parameters, which is equivalent to removing the bottom k2 parameters as k2+k1+k=1. Algorithm 1 does not show this directly, and we will revise this to give a more clear presentation.
>
>
>
> >  Q5: Assumption 1: $\gamma_1$ and $\gamma_2$ are constants determined by k1, k2, and k. What are their explicit definitions? Please write them out because they are used in main theorems and lemmas and should not be undefined.
>
>
>
> R5:  In Assumption 1, we use $\gamma_1$, $\gamma_2$ to denote upper bound of $||\nabla \mathbf{W} -
> \textnormal{DGP} (\nabla \mathbf{W}) ||_2$ and the lower bound of $||\nabla \mathbf{W} -\textnormal{ADGP}(\nabla \mathbf{W}) ||_2$. That said, the discarded gradients elements via DGP or ADGP are both norm-bounded (lower bounded by $\gamma_1  ||\nabla \mathbf{W}||^2$, and upper bounded by $\gamma_2  ||\nabla \mathbf{W}||^2$).
> We will make them more clear in the revised paper.
>
>
>
> >  Q6.1 :The comparison of DP with other defenses are not apple-to-apple because the DP accuracy is much lower.
>
>
> R6.1: Due to limited time, we have not yet found the setting where DP achieves the best accuracy. So far, we find using Gaussian noise with a variance of 0.01 DP performs much better in terms of the accuracy, and the corresponding results are shown in the tables below.
>
>
>
> Privacy evaluation under rob attack.
>
> |       | DGP      | DP     |
> | ----- | -------- | ------ |
> |       | Cifar100 |        |
> | LPIPS | 0.482    | 0.39   |
> | SSIM  | 0.078    | 0.2439 |
> |       | Imagenet |        |
> | LPIPS | 0.918    | 1.31   |
> | SSIM  | 0.022    | 0.2103 |
>
>
>
> Accuracy evaluation on Cifar dataset with LeNet(Zhu)
>
> |          | Baseline | DP     | ADGP   |
> | -------- | -------- | ------ | ------ |
> | CIFAR10  | 59.66%   | 56.46% | 58.98% |
> | CIFAR100 | 32.47%   | 28.89% | 30.39% |

---

### Official Review · Reviewer_jpQC · 2022-07-12

**Rating:** 4
**Confidence:** 5
**Soundness:** 2 fair
**Presentation:** 3 good
**Contribution:** 3 good

**Summary:**

The paper proposes a new gradient pruning strategy called "Aligned Dual Gradient Pruning" for federated learning (FL). It differs from the standard pruning approach of top-k sparsification in two ways - (i) instead of retaining the top-k gradients, it removes both top-k1 and bottom-k2 gradients, thereby increasing the distance between the original gradients and the exchanged gradients, and (ii) one of the clients is allowed to broadcast the locations of its top-2k gradients, so that all the other clients can choose from this list of locations while picking the gradients to transmit.

**Questions:**

1) What is the contribution of the error accumulation step? How will the proposed method behave if error accumulation is removed?

2) If the server keeps an account of the gradients transmitted by the client in each round as well as the locations of the top-2k gradients, can the server recover almost the full gradient of the clients over a few rounds of communication?

3) How will the proposed alignment scheme work in the presence of non-iid data?

4) What is the privacy impact of the alignment step? Will it leak more information than top-k sparsification since the server knows the locations of the top-2k gradients? Can more sophisticated attacks be designed leveraging this information?

5) What is the impact of the hyperparameters (k1, k2, and k) on convergence, accuracy, and privacy?



**Limitations:**

The paper discusses one limitation related to the assumption that all the clients need to be honest for the proposed scheme to work. But it overestimates the privacy benefits of the proposed approach.

**Strengths And Weaknesses:**

Strengths:

1) A variant of the top-k gradient pruning strategy has been proposed to enhance robustness against gradient inversion attacks (GIAs).
2) Both theoretical results and experiments have been presented to demonstrate that the proposed method works.

Weaknesses:

1) It is fairly obvious that if the top-k1 gradients are suppressed, it will lead to less information leakage compared to top-k sparsification. However, it is hard to believe that it will have no effect both on the convergence speed and final accuracy. Though the paper provides a theoretical result (Theorem 2) and experimental data (Table 2 and Figure 4), a concrete intuitive explanation of this unexpected behavior is required. Is this solely due to the error accumulation step? How will the proposed method behave if error accumulation is removed?

2)  There is no mention about non-iid data in the whole paper. One would expect that the proposed alignment strategy will completely breakdown in the case of non-iid data, because there will be very few intersections between the top gradients of different clients. A careful analysis of this issue is clearly necessary.

3) The most critical weakness of the paper is the lack of any analysis about the privacy leakage introduced by the alignment step. Since one of the clients transmits the binary matrix $\mathcal{I}$, it is trivial for the server to know the locations of the top 2k gradients of this client. Even if the client does not transmit the top-k1 gradient values, the server knows that locations with value 1 in $\mathcal{I}$ must have values greater than or comparable to that of the gradients that are actually transmitted by the client. Wouldn't this leak more information about the client's gradients compared to even the top-k sparsification strategy? Does any of the attack techniques considered in this paper exploit this knowledge?

4) There is no ablation study to evaluate the impact of the various components, especially alignment step and error accumulation step.

---

> ### Author Response · Authors · 2022-08-02
> **Response to Reviewer jpQC (Part 1)**
>
> Thanks for your thoughtful comments. We provide the following responses for your concerns.
>
>
>
> >  W1: It is fairly obvious that if the top-k1 gradients are suppressed, it will lead to less information leakage compared to top-k sparsification. However, it is hard to believe that it will have no effect both on the convergence speed and final accuracy. ?
>
>
> Re: To best explain our intuition, we start with DGP (suppressing both top $k_1$ and bottom $k_2$ elements). On the client side, each client has his own top $k_1$ and bottom $k_2$ set, so the convergence and final accuracy will be still (roughly) the same as with Fed-SGD. For example, for a particular user $i \in [1, N]$, his discarded information will be (high-likely) provided by other users $i' \neq i$ in the next round of iteration. Note that according to Assumption 3, each users' unpruned gradients is an unbiased estimation of the true central gradients, which makes DGP convergence roughly the same as of Fed-SGD.
>
> The intuition of ADGP follows the same reasons as above, with the only exception that in each round, all users send their respective gradient elements within the same location set (represented by the binary matrix $\mathcal{I}$) to the server. If the location binary matrix $\mathcal{I}$ remains the same for all iteration rounds, then clearly our method will fail with certainty in both converge and accuracy (as the reviewer pointed out). However, the location matrix $\mathcal{I}$ is randomly generated for each round (Algorithm~2 in the draft), thus avoiding this drawback naturally. Moreover, the error accumulation step indeed prompts the diversity of $\mathcal{I}$ among different iteration rounds, thus reducing the impact of gradient pruning on the converge and accuracy further.
>
> Theoretically, ADGP has a negative impact on the convergence and accuracy of the model. As shown in Theorem 2, compared to baseline Fed-SGD, ADGP introduces a negative term 4$\eta^2K^2\frac{\gamma_2}{2}(\frac{2+\gamma_2}{1-\gamma_2})^2(G^2+\sigma ^2)+2K\eta(G^2+\frac{\sigma^2}{N})$.
>
> Here, we further give some experimental results to validate the role of the error feedback mechanism. Experiments with larger models under different settings can be provided with more time.
>
> For ease of expression, we use EF$=$T to represent ADGP with error feedback, EF$=$F to represent ADGP without error feedback.
> The non-IID experiments are performed using LeNet(Zhu) on MNIST and ResNet18 on CIFAR10. We follow the data partition method proposed in FLtrust [1] to create  heterogeneous data distribution and set the non-IID  degree to be $q=0.3$ and $q=0.5$. The results are as follows.
>
> Ablation study of error feedback on IID data with LeNet(zhu).
>
>
>
> |          | Baseline | EF=T   | EF=F   |
> | -------- | -------- | ------ | ------ |
> | CIFAR10  | 59.66%   | 58.98% | 46.93% |
> | CIFAR100 | 32.47%   | 30.39% | 13.30% |
>
> Ablation study of error feedback on non-IID with $q=0.3$.
>
> |         | Baseline | EF=T   | EF=F   |
> | ------- | -------- | ------ | ------ |
> | MNIST   | 95.85%   | 94.94% | 94.51% |
> | CIFAR10 | 75.35%    | 68.63% | 67.13% |
>
>
>
> Ablation study of error feedback on non-IID with  $q=0.5$.
>
> |         | Baseline | EF=T   | EF=F   |
> | ------- | -------- | ------ | ------ |
> | MNIST   | 95.47%   | 93.64% | 86.55% |
> | CIFAR10 | 55.04%   | 49.75% | 46.98% |
>
>
>
> As validated from the above experimental results, error feedback boosts the performance of ADGP under the IID setting (about $12% - 17%$ accuracy improvement), while it provides very limited benefits to the non-IID setting (generally $\le 1%$ accuracy improvement). This verifies our conlucsion above, the error accumulation improves the diversity of $\mathcal{I}$ among different iteration rounds. In the non-IID case, the diversity of $\mathcal{I}$ is already high by nature.

---

> > ### Author Response · Authors · 2022-08-02
> > **Response to Reviewer jpQC (Part 2)**
> >
> > >  proposed alignment strategy will completely breakdown in the case of non-IID data, because there will be very few intersections between the top gradients of different clients.
> >
> >
> >
> > Re: The non-IID setting has limited impact on the alignment strategy since our alignment step is a position-based (rather than a value-based) filtering operation, in a round-by-round manner. Specifically, during alignment, a randomly chosen user first constructs a location matrix $\mathcal{I}$ and broadcasts it to the rest of the clients, each of which uploads $k$ gradients within this location matrix. Thus, in non-IID setting, although the values of gradients vary significantly among users, each uploaded gradient will still stay within the same location matrix $\mathcal{I}$ and users only need to download global gradients' parameters associated with $\mathcal{I}$ after aggregation. Furthermore, we set the size of this location matrix to be $2k$ and let each user select $k$ gradients, which are the largest among these $2k$ locations (indicated by $\mathcal{I}$), thus mitigating the impact of the alignment step to a certain extent.
> >
> >
> > Here we give the ablation study of alignment strategy to empirically validate our viewpoint.
> >
> >
> >
> >  Ablation study of alignment strategy on IID data with LeNet(zhu).
> >
> > |          | Baseline | DGP    | ADGP   |
> > | -------- | -------- | ------ | ------ |
> > | CIFAR10  | 59.66%   | 59.22% | 58.98%  |
> > | CIFAR100 | 32.47%   | 30.94% | 30.39% |
> >
> >
> >
> > Ablation study of alignment strategy on non-IID data with q=0.3.
> >
> > |         | Baseline | DGP    | ADGP   |
> > | ------- | -------- | ------ | ------ |
> > | MNIST   | 95.85%   | 95.45% | 94.94% |
> > | CIFAR10 | 75.35%   | 69.73% | 68.63% |
> >
> >
> >
> >  Ablation study of alignment strategy on non-IID data with $q=0.5$.
> >
> > |         | Baseline | DGP    | ADGP   |
> > | ------- | -------- | ------ | ------ |
> > | MNIST   | 95.47%   | 93.89% | 93.64% |
> > | CIFAR10 | 55.04%   | 50.99% | 49.75% |
> >
> > It is clear from these tables that the performance differences of DGP (without alignment) and ADGP (with location alignment) are quite small for all evaluated cases.
> >
> >
> >
> > >  W3: The most critical weakness of the paper is the lack of any analysis about the privacy leakage introduced by the alignment step.
> >
> >
> >
> > R3: Thanks for your comments about the side-channel leakage of our protocol ADGP. And indeed, we take this leakage into consideration in our design. Put simply,
> > the server cannot infer any information about a specific user during an iteration round. We analyse this fact by showing that the view of the server on user $i$ and user $j$ ($i \neq j, i, j \in [1, N]$) are exactly the same, which actually disables the attacker mentioned by the reviewer.
> >
> >    1. In the iteration round $t$, a user $i_0$ is randomly selected and user $i_0$ will generate the binary location matrix $\mathcal{I}$ according to his top-$2k$ gradients elements location set $\mathcal{T}_{2k}$.
> >    2. This $\mathcal{I}$ is send to all the $N-1$ users, but not the server (discussed in lines 177-178, Sec.~4.3  in the draft).
> >    3. All the users will set the top-$k_1$ gradient elements in the location set $\mathcal{T}_{k_1}$ as $0$, including the user $i_0$. All the gradients are updated here.
> >   4. All the users will then send the $k$ largest gradient elements that belong to $\mathcal{I}$ to the server. In real implementation, this can be done by letting all users to send $2k$ locations, $k$ largest gradient elements with their associated locations to the server (to save bandwidth if the size of $\mathcal{I}$ is too large).
> >
> >
> > At the server side, it receives $N$ copies of $2k$ locations (all are the same for all $N$ users), and $k$ largest gradient elements with their associated locations for each user. Concerning with user $i$ and user $j$ ($i \neq j,  i, j \in [1, N]$), the views at the server side for these two users are exactly the same: $2k$ locations and $k$ largest gradient elements with associated locations.
> > The server's view on user $i$'s update and its view on user $j$'s cannot be distinguished.
> > As such, if all the users follow the steps specified in ADGP faithfully, the leakage will be eliminated by nature.
> >
> > We will add security analysis for the alignment step in our revised paper.

---

> > > ### Author Response · Authors · 2022-08-02
> > > **Response to Reviewer jpQC (Part 3)**
> > >
> > > >  Q2: If the server keeps an account of the gradients transmitted by the client in each round as well as the locations of the top-2k gradients, can the server recover almost the full gradient of the clients over a few rounds of communication?
> > >
> > > Re: The direct answer to this concern is no. The server's inference on the full clean gradient of the client can be only launched from two different aspects: 1) infer a full clean gradient from the suppressed gradient uploaded by the user; 2) infer a full clean gradient from the aggregated gradient with locations used in each round.
> > >
> > > Clearly, case 1) will lead to a failure by certain due to information loss.
> > > For case 2), assume $t$ rounds of communication have happened and the attacker knows all the location matrices $\mathcal{I}^1$, $\mathcal{I}^2$, $\cdots$, $\mathcal{I}^t$.
> > > The full clean local gradients are $g_i^1, g_i^2, \cdots,  g_i^t$, while the server's observations are just $\textnormal{ADGP}(g_i^1), \textnormal{ADGP}(g_i^2), \cdots, \textnormal{ADGP}(g_i^t)$ for all $i \in [1, N]$ users. We then proceed with (proof by) contradiction.
> > >
> > > Say if the server is able to recover any $g_i^j$ (full clean gradient of client $i$ after $j$ iterations), then he must be able to recover the full clean gradient $g_{i'}^j$ of another user $i'$ ($i' \neq i$) by using the same function. Call this function to all the $N$ users, the server will have all $g_1^j, g_2^j, \cdots,  g_N^j$. Now he can easily determine who produces the $\mathcal{I}^j$ (generated by a random user at the iteration $j$) by comparing all the full clean local gradients against all the suppressed local gradients uploaded by $N$ users.
> > > That said, the server breaks the random selection mechanism (cryptographic secure random number generator), which is impossible.
> > >
> > > Finally, the only exception of our analysis is that when the learning process starts to converge, all the gradients, either clean local gradients, suppressed gradients or even aggregated gradients, converge to all zeros. The attacker can get this constant vector by the nature of learning.
> > >
> > >
> > >
> > >
> > >
> > >
> > > >  Q5:What is the impact of the hyperparameters ($k_1$, $k_2$, and $k$) on convergence, accuracy, and privacy?
> > >
> > > Re A large $k$ will lead to a bad convergence and accuracy of the model. When $k$ is fixed, a larger $k_1$ makes a smaller $k_2$, and the convergence and accuracy will be worse, while the privacy protection will be strengthen. In the supplementary, we provide related experiments in section A.2.5.
> > >
> > >
> > >
> > > References
> > >
> > > [1] Cao, Xiaoyu, et al. "FLTrust: Byzantine-robust Federated Learning via Trust Bootstrapping." Proceedings of NDSS. 2021.

---

> > > ### Comment · Reviewer_jpQC · 2022-08-08
> > > **Security Analysis**
> > >
> > > 1) Thanks for the responses about the non-iid scenario. I still think the argument about diversity is somewhat fuzzy, but I'm willing to accept it.
> > >
> > > 2) The error feedback mechanism does appear to have some impact on the accuracy. It would also be interesting to study the impact of the error feedback mechanism on the convergence rate.
> > >
> > > 3) My main concern about the lack of clear security/privacy analysis (quantifying the side channel information leakage) remains unresolved. The key assumption in Step 2 is that the server does not know the binary location matrix. This is an unrealistic and strong assumption. If clients can communicate directly without the knowledge of server, a number of problems in FL can be easily solved. Also, the proof by contradiction presented below is not convincing. The question is not about whether the server can recover the complete clean gradient of each client. The real issue is whether the server can get sufficient information to "approximately recover" the clients' data.

---

> > > > ### Author Response · Authors · 2022-08-09
> > > > **Further Response to Reviewer jpQC**
> > > >
> > > > Thank you for your kind response to our reply. We fully agree with your point that the non-IID scenario is hard to quantify
> > > >  (*somewhat fuzzy*), and we will try our best to make it clear in the final version. Moreover, we will also add the ablation study of the error feedback mechanism on the convergence rate to the supplementary (due to space limit in the main text).
> > > >
> > > > For the security and privacy analysis of ADGP, it is true that the current version is built on a strong assumption (a random client can broadcast the location to the other clients) and we don't have a meaningful way to quantify the side channel information leakage. However, we still think that our most important contribution, DGP, whose security does not depend on this assumption, is not affected by this side-channel attack. Indeed, we provide a meaningful theoretical guarantee for DGP (only regarding optimization-based attacks as shown in Definition 1). And we are aware that there might still be other side channel leakages (like time- and energy-based side channels) for DGP, but this is not the focus of this work. In view of this, we will remove ADGP from the security analysis, and claim it as a pure communication efficiency improvement (with unknown side channel information leakage for us).
> > > >
> > > > Thanks again for pointing out this important problem to us.

---

### Official Review · Reviewer_XcaE · 2022-07-14

**Rating:** 7
**Confidence:** 3
**Soundness:** 3 good
**Presentation:** 3 good
**Contribution:** 3 good

**Summary:**

- The paper proposes a defense against Gradient Inversion Attacks (where input images can be reconstructed by an attacker from the public gradients).
- The crux of the proposed approach is (a) pruning gradients: where top-$k_1$ and bottom-$k_2$ parameters are pruned and (b) gradient-location bounding: such that users across FL prune similarly which results in sparsified models even at download-time.
- The paper is also accompanied with theoretical insights validating the choices e.g., perturbed vs. non-perturbed gradients are proportional to reconstruction errors.
- The defense approach is evaluated against three attacks (Rob, IVG, R-GAP) and two datasets (CIFAR10, CIFAR100). Evaluation indicates a performance improvement over existing defense against all attacks, while retaining the utility (efficiency, accuracy).

**Questions:**

Please see the questions under concerns.

**Limitations:**

Yes, the authors have adequately discussed the limitations in the main paper.

**Strengths And Weaknesses:**

### Strengths
1. Insight: The core insight is intuitive. Existing top-k approaches *retains* the top-k parameters and the authors show that this results in keeping the sparsified gradient close to the original and thereby increasing the attack effectiveness (Proposition 1). In contrast, the approach *prunes* the top-k gradients and thereby lowering the reconstruction performance.
2. Evaluation: The authors benchmark the approach against multiple classes of GIA attacks (analytical, optimization-based, ...).
3. Results: The results are significant and promising. The approach while almost perfectly retaining the target model accuracy (93.17 vs. 93.44) thwarts all considered classes of attacks.

### Concerns
1. Gradients attacked from which iteration?: The evaluation methodology is slightly unclear. Specifically, it is unclear the gradients from which training iteration is used to demonstrate system-under-attack performance. Are the authors aware whether this has an impact to attack performance? I reckon the gradients at the early training iterations are more
2. Choice of attacks: While I appreciate the authors consider different classes of attacks, some recent popular attacks seem to have been left-out e.g., GradInversion [26]. Specifically, the class of optimization-based attacks which additionally regularize fidelity.
3. Some evalution details unclear: While the paper indicates evaluation on 2x CIFAR10 models and CIFAR100, it is often not clear for which model/dataset the results are. For instance, the setting is unclear in Fig.1 and Table 2 (which I think is CIFAR10?), but Fig. 2a displays attacks on CIFAR100.

---

> ### Author Response · Authors · 2022-08-02
> **Response to Reviewer XcaE**
>
> Thanks for your thoughtful comments. We provide the following responses for your concerns.
>
>
>
> > W1: Gradients attacked from which iteration? The evaluation methodology is slightly unclear.
>
>
>
> Re: Keeping in pace with existing works like Soteria [10], ATS [11], etc., we make the attacker start the attack from the first iteration.
>
>
>
> > W2: Choice of attacks: While I appreciate the authors consider different classes of attacks, some recent popular attacks seem to have been left-out e.g., GradInversion [26]. Specifically, the class of optimization-based attacks which additionally regularize fidelity.
>
>
>
> Re: We have left out GradInversion [26] in the comparison because it performs much worse than other schemes during our experiments, although it is a recently proposed work. Here we provide some results evaluated with ResNet18 on imagenet, as shown below.
>
>
>
>  The evaluation of DGP's defense against GradInversion [26] with different batchsize.
>
>
>
> |       | batchsize=1 | batchsize=4 | batchsize=8 |
> | ----- | ----------- | ----------- | ----------- |
> | LPIPS | 0.66        | 0.68        | 0.68        |
> | SSIM  | 0.283       | 0.2898      | 0.2853      |
>
>
>
>
>
>
>
> >W3: Some evaluation details unclear: While the paper indicates evaluation on 2x CIFAR10 models and CIFAR100, it is often not clear for which model/dataset the results are.
>
>
>
> R3: As illustrated in our experimental setup, three  attacks are implemented strictly following the original settings, thus resulting in using different models and datasets. In both Fig. 1 and Table 2, IVG and R-GAP are evaluated on CIFAR10, while Rob is evaluated on CIFAR100. In Fig. 2(a), we separately evaluate IVG attack with different settings to demonstrate the the advantage of our DGP compared with Top-k. We will make it more clear our revised paper.

---

### Official Review · Reviewer_MAQA · 2022-07-16

**Rating:** 4
**Confidence:** 3
**Soundness:** 2 fair
**Presentation:** 3 good
**Contribution:** 2 fair

**Summary:**

This paper considers an important and timely problem in Federated Learning, investigating defense mechanism against model inversion attack in terms of three key metrics: privacy, utility, and communication efficiency. To provide better trade-off among the three key metrics, this paper firstly proposes a simple sparsification technique, named dual gradient pruning (DGP), which removes top-k1 largest values and bottom-k2 smallest values in local model parameters. In the second component, named gradient location bounding, communication cost is further reduced by ensuring that the sparsified region of different users are the same.
This paper theoretically shows that proposed scheme guarantees better privacy against model inversion attack. In addition, convergence analysis of the proposed method is provided with IID dataset distribution (i.e., local computation of each user is unbiased to the full gradient) and assumption on DGP.


**Questions:**

1.	How can we represent gamma_1 and gamma_2 with respect to k_1, k_2, and k?
2.	In experiments, what is N (number of users), and how the dataset (CIFAR10/100) is distributed over N users?


**Limitations:**

As this paper considers the privacy in FL, it has no negative societal impact.

**Strengths And Weaknesses:**

Overall, this paper is well structured and written, and hence easy to follow. Intuition behind the motivation of DGP/ADGP is well described in the section 4. While the proposed pruning technique is very simple, it can incur further research investigating privacy and utility trade-off in sparsification/pruning schemes in FL. Especially, combining sparsification and other defense mechanism (DP, secure aggregation, etc) would be very interesting future direction. It also includes extensive experiments with various attack models and defense mechanisms to demonstrate the stronger privacy guarantee and comparable utility of the proposed scheme.

I have the following concern/comment:

1. In theoretical analysis, it seems that this paper relies on strong and unrealistic assumptions. This paper considers IID data distribution (i.e., in assumption 3, local stochastic computation is unbiased to the full gradient), which is not the case in FL. In addition, as the assumption 1 is very important for the analysis of convergence, more justification/explanation is needed. For instance, while k1 is an important parameter which determines the utility (and convergence performance), relationship between \gamma_2 and k_1 is not presented.

2. In section 4, the role of e_i^t is not described. We can know that it is accumulated error of user i at round t in section 5 (after reading the lemma 1, but it is helpful to understand the proposed scheme if explanation will be included in section 4.

---

> ### Author Response · Authors · 2022-08-02
> **Response to Reviewer MAQA (Part 1)**
>
> Thanks for your valuable comments. Please find our responses below for your questions.
>
> > *1. In theoretical analysis, it seems that this paper relies on strong and unrealistic assumptions. This paper considers IID data distribution (i.e., in assumption 3, local stochastic computation is unbiased to the full gradient), which is not the case in FL.
>
> Re: First, in Assumption 3, we assume that stochastic gradients are the unbiased estimation of the full gradient. This assumption does not require the IID data distribution, and it can still be established in the non-IID scenario whenever federated learning is still valid. In particular, consider the basic Fed-SGD aggregation that all the clients' local gradient
>
> $$\Delta \mathbf{W}_i^t~(i=[1,N])$$
>
> are averaged to obtain the aggregated gradient
> $$\nabla l(\mathbf{W}^t)$$
> at the central server. As
> $$t \rightarrow \infty$$
>  and the central model start to converge, it must be true that
> $$\lim_{t \rightarrow \infty} \left(\nabla \mathbf{W}_i^\infty - \nabla l(\mathbf{W}^\infty) \right)= 0$$
> (otherwise, it is easy to deduce a contradiction that $$\nabla l(\mathbf{W}^\infty)$$ will not converge). In this sense, Assumption 3 is realistic for both IID and non-IID data distribution in the federated learning scenario.
> Note that our proposed methods DGP and ADGP are just variances of Fed-SGD, our converge analysis (Theorem 2) proves that our methods converge similar to Fed-SGD. Last-but-not-least, similar assumption used in this paper can be found in [1-2].
>
>
> 2. In addition, as the assumption 1 is very important for the analysis of convergence, more justification/explanation is needed. For instance, while k1 is an important parameter which determines the utility (and convergence performance), relationship between $\gamma_2$ and $k_1$ is not presented.
>
> We use $\gamma_1$,$\gamma_2$ to denote upper bound of\;$||\nabla \mathbf{W} -{DGP} \nabla \mathbf{W}) ||_2$/$|||\nabla \mathbf{W}||_2$ and the lower bound of $||\nabla \mathbf{W} -{ADGP}(\nabla \mathbf{W}) ||_2$/$||\nabla \mathbf{W}||_2$. The following is an analysis of the relationship between these parameters.
>
>
> > *W2: In section 4, the role of $e_i^t$ is not described. We can know that it is accumulated error of user i at round t in section 5 (after reading the lemma 1, but it is helpful to understand the proposed scheme if explanation will be included in section 4.*
>
>
>
> Re: Thanks for the useful suggestion. We will make changes in the revised version.
>
>
>
>
>
>
> Abuse the notion of $k_1$, $k_2$ and $k$ to denote ratio (e.g., $k_1/{Size}(\nabla \mathbf{W})$), the relationship between $\gamma_1$, $\gamma_2$ and $k_1$, $k_2$, $k$ are as follows:
> $$
> \gamma _1=(1-\sqrt{1-k_1} )^2 ,  (1-\sqrt{2k-k_1} )^2\le\gamma _2
> $$
>
>
>  Proof:
>
> To simplify the expression, we use $x$ to denote the gradient and $||\cdot||$ denotes $||\cdot||_2$. [4] states the following property of topl$(x)$ (i.e., retain the top $l$-ratio of $x$):
> $$
> ||x - {top}l(x)|| \leq \sqrt{1-l} ||x|| \tag{1}
> $$
> According to formula (1), it is easy to obtain formula (2):
> $$
> ||\{top}l(x)|| \ge(1-\sqrt{1-l} )||x|| \tag{2}
> $$
> Then, we can show the relationship between $\gamma_1$ and $k_1$, $k2$, $k$ as follows:
> $$
> ||x-{DGP}(x)||=||x-(x-{top}(k_1)(x)- {bottom}(k_2)(x))||
> \overset{(a)}{\ge} ||{top}(k_1)(x)|| \overset{(b)}{\ge}(1-\sqrt{1-k_1} )||x|| \tag{3}
> $$
> where (a) is based on the properties of the 2-norm, and (b) is directly an application of (2).
>
> Therefore, we set $\gamma _1=(1-\sqrt{1-k_1} )^2$ to satisfy:
> $$
> \sqrt{\gamma_1 } ||x|| \le ||x- {DGP}(x)||
> $$
> Next,  we  prove the relationship between $\gamma_2$ and $k_1$, $k$
>
> We set $\gamma_2$ that satisfies:
> $$
> ||x-ADGP(x)||\le\sqrt{\gamma _2} ||x||\tag{4}
> $$
>
>
> Because ADGP is random, we consider the worst case, i.e., $bottom(2k-k1)(x)$  and $topk1(x)$ are selected by the matrix, the user can only send $bottom(2k-k1)(x)-bottom(k-k1)(x)$. Then, we have:
> $$
> ||x-ADGP(x)||=||x-(bottom(2k-k1)(x)-bottom(k-k1)(x))||\ge ||x-bottom(2k-k1)(x)||$$
> $$=||Top(1-2k+k1)(x)||\overset{(c)}{\ge}(1-\sqrt{1-(1-2k+k1)} )||x||=(1-\sqrt{2k-k1} )||x||\tag{5}
> $$
> where (c) is directly an application of (2).
>
> Then, we have:
> $$
> (1-\sqrt{2k-k1} )^2\overset{(d)}{\le}\gamma_2
> $$
> where (d) is based on (4) and (5).

---

> > ### Author Response · Authors · 2022-08-02
> > **Response to Reviewer MAQA (Part 2)**
> >
> > > Q2 : In experiments, what is N (number of users), and how the dataset (CIFAR10/100) is distributed over N users?
> >
> >
> >
> > Re: As stated in Sec. 6.3 and experimental details in the supplementary, the number of participating users is N=10, and our experiments are conducted on the balanced datasets (ie.,  i.i.d setting). For non-i.i.d scenarios, here we add experiments to demonstrate the effectiveness of our scheme.
> >
> > The experiments are performed using LeNet(Zhu) on MNIST and ResNet18 on CIFAR10. We follow the data partition method proposed in FLtrust[3] to create  heterogeneous data distribution and set the non-i.d.d  degree to be q=0.3 and q=0.5.
> > The experiment results are shown below. We will provide more experiments with different datasets and models in the modified version.
> >
> >
> >
> > Accuracy results on non-i.i.d with q=0.3
> >
> > |         | Baseline | DGP    | ADGP   |
> > | ------- | -------- | ------ | ------ |
> > | MNIST   | 95.85%   | 95.45% | 94.94% |
> > | CIFAR10 | 75.35%   | 69.73% | 68.63% |
> >
> >  Accuracy results on non-i.i.d with q=0.5
> >
> > |         | Baseline | DGP    | ADGP   |
> > | ------- | -------- | ------ | ------ |
> > | MNIST   | 95.47%   | 93.89% | 93.64% |
> > | CIFAR10 | 55.04%   | 50.99% | 49.75% |
> >
> > References:
> >
> > [1] Avdiukhin, Dmitrii, and Shiva Kasiviswanathan. "Federated learning under arbitrary communication patterns." International Conference on Machine Learning. PMLR, 2021.
> >
> > [2]Girgis, Antonious, et al. "Shuffled model of differential privacy in federated learning." International Conference on Artificial Intelligence and Statistics. PMLR, 2021.
> >
> > [3] Cao, Xiaoyu, et al. "FLTrust: Byzantine-robust Federated Learning via Trust Bootstrapping." Proceedings of NDSS. 2021.
> >
> > [4] Alistarh, Dan, et al. "The convergence of sparsified gradient methods." *Advances in Neural Information Processing Systems* 31 (2018).

---

> > ### Comment · Reviewer_MAQA · 2022-08-03
> > **Further question about the unbiased assumption**
> >
> > Thanks for your response.
> >
> > I still have a question about the unbiased assumption.
> >
> > Let's say $\mathcal{D}$ is the full dataset and it is concatenation of local dataset $\mathcal{D}_i$ of user $i\in [N]$.
> >
> > Typically, unbiased estimator of local SGD means $\mathbb{E}_{X \sim \mathcal{D}_i} [g_i(W^t;X)] $ = $\nabla l(W^t; \mathcal{D}_i)$ where $g_i$ is the local stochastic gradient computation at user $i$ ($\nabla W_i ^t$ in your notation).
> >
> > In IID setting, this local SGD can be unbiased estimator for the full gradient, i.e.,
> > $\mathbb{E}_{X \sim \mathcal{D}_i } [g_i(W^t;X)]  $ = $ \nabla l(W^t; \mathcal{D})$.
> >
> > In non-IID setting, however, expected value of local SGD is not equal to the full gradient as $ \nabla l(W^t; \mathcal{D}_i) \neq \nabla l(W^t; \mathcal{D})$.
> >
> > Please correct me if I am missing something.

---

> > > ### Author Response · Authors · 2022-08-06
> > > **Further Response to Reviewer MAQA**
> > >
> > > Thanks for your feedback. We think this is a thought-provoking question. Before presenting our view on this question, we need to first point out this is a common assumption in existing federated learning or distributed learning literature studies [1-5] when proving convergence.
> > >
> > >
> > >
> > > We discuss the rationale of this assumption from the perspective of when the model starts to converge.
> > >
> > > Suppose there are $N$ users, corresponding to the splitted datasets  $D_1, D_2, \cdots, D_N$ (note that here the splitting can be either IID or non-IID), we think it is reasonable to assume
> > > $E_{x\sim D[N]\setminus i}(\nabla l(W^t,x))$ is very close to the unbiased estimate of $\nabla l(W^t, D)$. In other words, the gradients produced from $(D_1, \cdots D_{i-1}, D_{i+1}, \cdots, D_N)$ is an unbiased estimate of the full gradient $\nabla l(W^t, D)$. Indeed, this is true when $N$ becomes large, which is the case in real FL industrial applications ($N$ reaches to billions in certain applications).
> > >
> > > When the model converges, it must be true $E_{x\sim D[N]\setminus i}(\nabla l(W^t, x)) \approx \nabla l(W^t, D)=0.$
> > > Considering the fact $E_{x\sim D_i}(\nabla l(W^t,x)) +E_{x\sim D[N]\setminus i}(\nabla l(W^t,x))=\nabla l(W^t,D)= 0$, it is easy to see
> > > $E_{x\sim D_i}(\nabla l(W^t,x)) \approx E_{x\sim D[N]\setminus i}(\nabla l(W^t, x)) \approx \nabla l(W^t, D)=0.$
> > > So, theoretically speaking, the $i$-th user converges similarly as the rest when the central model indeed converges. This holds true regardless of the IID or non-IID data distribution assumption.
> > >
> > >
> > > But this theoretical convergence is difficult to achieve in reality. Indeed, when the local gradients sum up to (near) $0$, the central model is considered to be converged. But the user's own local gradient is not necessarily $0$ in this case. From some experimental results, we find that in the IID setting, the local gradients across users are similar (the same finding is also reported in research papers like [6]). So, if the aggregation result is (near) zero, similarity ensures almost zero local gradient updates. In contrast, for the non-IID setting, (near) zero aggregated central gradient can be just due to a small positive local update and a negative local update being cancelled with each other.
> > >
> > > For the above reasons and considering this assumption is widely used for theoretical convergence proofs, we think this is not a huge restriction on our work (note that we also only used it for convergence proof but strictly not for security analysis). We thank the reviewer again for pointing us to this thought-provoking problem and helping us understand the restrictions of federated learning and our work much better. **We sincerely expect the response clarifies the reviewer's concern.**
> > >
> > >
> > >
> > > References:
> > >
> > >
> > >
> > > [1] Chen, Chia-Yu, et al. "Scalecom: Scalable sparsified gradient compression for communication-efficient distributed training." Advances in Neural Information Processing Systems 33 (2020): 13551-13563.
> > >
> > > [2] Sohn, Jy-yong, et al. "Election coding for distributed learning: Protecting signsgd against byzantine attacks." Advances in Neural Information Processing Systems 33 (2020): 14615-14625.
> > >
> > > [3] Haddadpour, Farzin, et al. "Federated learning with compression: Unified analysis and sharp guarantees." International Conference on Artificial Intelligence and Statistics. PMLR, 2021.
> > >
> > > [4] Liu, Xiaorui, et al. "A double residual compression algorithm for efficient distributed learning." International Conference on Artificial Intelligence and Statistics. PMLR, 2020.
> > >
> > > [5] Yu, Yue, Jiaxiang Wu, and Longbo Huang. "Double quantization for communication-efficient distributed optimization." Advances in Neural Information Processing Systems 32 (2019).
> > >
> > > [6] Cao, Xiaoyu, et al. "FLtrust: Byzantine-robust federated learning via trust bootstrapping." NDSS (2021).

---

> > > > ### Comment · Reviewer_MAQA · 2022-08-08
> > > > **Follow-up question**
> > > >
> > > > Thanks again for your kind and detailed explanation.
> > > >
> > > > First, I'd like to point out that [1], [2] only consider iid data distribution. [3] considers both iid and noniid setting, and you can check the difference of iid/noniid distributions in Section 5.1 and 5.2 of [3]. For the noniid setting (i.e., heterogeneous setting), in Assumption 4, unbiased stochastic gradient means that expected value of local stochastic gradient of user j is equal to g_j (g_j means the true gradient of local objective function at user j), and g_i and g_j are not equal for different user i, j.
> > > > Commonly, non-iid FL assumes that local stochastic gradients of different user are bounded [3, 4].
> > > > More clearly, you can check that in page 5 of [4], $\nabla f_i (x^*) \neq \nabla f_j (x^*)$ where $x^*$ is the optimal point (even when the model is converged), and [4] says that each individual $\nabla f_i (x^*)$ may be far away from zero, which is opposite to your claim, $\mathbb{E}_{X\sim D_i} [\nabla l (W^*, X)] \approx 0$.
> > > >
> > > >
> > > > References:
> > > >
> > > > [1] Chen, Chia-Yu, et al. "Scalecom: Scalable sparsified gradient compression for communication-efficient distributed training." Advances in Neural Information Processing Systems 33 (2020): 13551-13563.
> > > >
> > > > [2] Sohn, Jy-yong, et al. "Election coding for distributed learning: Protecting signsgd against byzantine attacks." Advances in Neural Information Processing Systems 33 (2020): 14615-14625.
> > > >
> > > > [3] Haddadpour, Farzin, et al. "Federated learning with compression: Unified analysis and sharp guarantees." International Conference on Artificial Intelligence and Statistics. PMLR, 2021.
> > > >
> > > > [4] Liu, Xiaorui, et al. "A double residual compression algorithm for efficient distributed learning." International Conference on Artificial Intelligence and Statistics. PMLR, 2020.

---

> > > > > ### Author Response · Authors · 2022-08-09
> > > > > **Further Response to Reviewer MAQA: Follow-up question**
> > > > >
> > > > > Thank you for the response and pointing out this important misunderstanding in our assumption. As per your kind suggestion, we carefully read the model convergence proof that appeared in [3] and [4], and confirmed that we can use the same assumption for convergence proof in our work (without relying on the unbiased assumption).
> > > > >
> > > > > In section A.1 of the supplementary material, we used the original unbiased assumption only in inequalities (7-8) to ensure that the magnitude of the stochastic gradient for each user is bounded. The bounded requirement can be easily achieved by the same assumptions used in [3] and [4]. Similar to [3, 4] and using the condition that the user's local stochastic gradient are bounded, we can also prove final convergence result after adjusting the learning rate.
> > > > >
> > > > > Thanks again for pointing out the mistake for us, and we will correct this in the revised version.
> > > > >
> > > > >
> > > > > References:
> > > > >
> > > > > [1] Chen, Chia-Yu, et al. "Scalecom: Scalable sparsified gradient compression for communication-efficient distributed training." Advances in Neural Information Processing Systems 33 (2020): 13551-13563.
> > > > >
> > > > > [2] Sohn, Jy-yong, et al. "Election coding for distributed learning: Protecting signsgd against byzantine attacks." Advances in Neural Information Processing Systems 33 (2020): 14615-14625.
> > > > >
> > > > > [3] Haddadpour, Farzin, et al. "Federated learning with compression: Unified analysis and sharp guarantees." International Conference on Artificial Intelligence and Statistics. PMLR, 2021.
> > > > >
> > > > > [4] Liu, Xiaorui, et al. "A double residual compression algorithm for efficient distributed learning." International Conference on Artificial Intelligence and Statistics. PMLR, 2020.

---

### Official Review · Reviewer_cSmE · 2022-07-28

**Rating:** 3
**Confidence:** 4
**Soundness:** 1 poor
**Presentation:** 3 good
**Contribution:** 3 good

**Summary:**

This paper proposes Aligned Dual Gradient Pruning (ADGP) as a defense against gradient inversion attacks.

A gradient inversion attack takes the gradient $\nabla W (x)$ (and the model $W$) and attempts to reconstruct the input data $x$.

Standard gradient pruning takes the gradient $\nabla W (x)$ and zeroes out the smallest coordinates. Intuitively, this makes it harder for the attacker as they are not given as much information.

Dual Gradient Pruning (DGP) zeroes out not only the smallest coordinates but also the largest coordinates, leaving only the "median" coordinates of the gradient.

ADGP modifies DGP by ensuring that all the participants zero out a similar set of coordinates, rather that independently choosing which coordinates to zero out.

The paper provides both experimental and theoretical analysis of ADGP. It give theoretical results that bound the error of a potential attacker. And it also gives experimental results where attacks are run against ADGP. This includes a comparison to other defenses and a utility analysis.

**Questions:**

I have a question about the theoretical results (Proposition 1) and the experimental results (Figures 1 & 2).

What is the optimization problem that the attacker is solving?

Abusing notation slightly, I can see two possible ways to do this: First, $$\arg\min_{x'} \\|\nabla W(x') - \mathsf{ADGP}(\nabla W(x))\\|$$ and, second, $$\arg\min_{x'} \\| \mathsf{ADGP}(\nabla W(x')) - \mathsf{ADGP}(\nabla W(x))\\|.$$
Which of these two is being considered in the paper? or is it something else?

Intuitively, my question is how does the attacker account for the ADGP defense?

I also have a minor question about Definition 1. There is a nested probability and expectation. What is the source of randomness for each of these?

---

The discussion with the authors clarified the question about what optimization the attacker is performing, namely $$\arg\min_{x'} \\|\nabla W(x') - \mathsf{ADGP}(\nabla W(x))\\|.$$
Unfortunately, this is the wrong way to formulate the problem. Intuitively, this assumes the attacker is not aware of the ADGP defence being used and applies a naive strategy. More precisely, the problem is that this objective is not minimized by $x'=x$ -- i.e., this objective does not lead to successful reconstruction.

This issue essentially invalidates the experimental and theoretical results. They assume a naive attacker and tell us nothing about what an attack tailored to this defence would do.

In the discussion I presented a simple counterexample to Theorem 1 in the paper. That is, the loss $\ell(w,x) = (x-\sum_i w_i)^2$, where $x$ is a scalar and $w$ is a vector. Given any single coordinate of the gradient and the value of $w$, it is possible to reconstruct $x$. ADGP does not protect against this attack.

The experimental results attempt "state-of-the-art" attacks. The problem is that an attack is tailored to a specific system. A state-of-the-art attack for one system may not be state of the art for another system.

Unfortunately, given this major flaw, I think the paper cannot be accepted.

**Limitations:**

Limitations are discussed

**Strengths And Weaknesses:**

Strengths:
 - This paper studies an important problem.
 - It proposes a simple and novel approach to protecting against gradient inversion.
 - Both theoretical guarantees and empirical evaluation are included.


Weaknesses:
 - The theoretical results are somewhat difficult to interpret. E.g. the parameters $\gamma_1$ and $\gamma_2$ from Assumption 1 appear in Theorem 1, but it is not clear what these parameters are and, hence, how to interpret the guarantee.
 - The experimental results do not consider varying the parameters $k_1$ and $k_2$, namely the number of large and small coordinates to zero out. Similarly the other defenses being compared against also have parameters that could be varied.
 - Some details about the results are unclear (see below).

---

> ### Author Response · Authors · 2022-08-02
> **Response to Reviewer cSmE**
>
> Thanks for your valuable comments. Please find our responses below for your questions.
>
>
>
>
>
> > W1: The theoretical results are somewhat difficult to interpret. E.g. the parameters and from Assumption 1 appear in Theorem 1, but it is not clear what these parameters are and, hence, how to interpret the guarantee.
>
> Re:
>
> 1. how to interpret these parameters.
>
>  In Assumption 1, we use $\gamma_1$, $\gamma_2$ to denote upper bound of $||\nabla \mathbf{W} -
> \textnormal{DGP} (\nabla \mathbf{W}) ||_2$ and the lower bound of $||\nabla \mathbf{W} -\textnormal{ADGP}(\nabla \mathbf{W}) ||_2$. That said, the discarded gradients elements via DGP or ADGP are both norm-bounded (lower bounded by $\gamma_1  ||\nabla \mathbf{W}||^2$, and upper bounded by $\gamma_2  ||\nabla \mathbf{W}||^2$).
> We will make them more clear in the revised paper.
>
>
>
> 2. how to interpret the guarantee.
>
> In Theorem 1, the assumptions made in Assumption 1 are used to analyze the security of our design according to Definition 1. As we remarked in the manuscript (line 112), a smaller $\epsilon$ indicates a better attack result when $\delta$ is given. Our Theorem 1 is aligned with Definition 1 in the sense that, after DGP protection, all attacks will only be able to achieve a worse ($\epsilon + \sqrt{\gamma_2} ||\nabla \mathbf{W}||_2  > \epsilon$) attack for the same $\delta$.
>
>
>
> > *W2: The experimental results do not consider varying the parameters and, namely the number of large and small coordinates to zero out. Similarly the other defenses being compared against also have parameters that could be varied.
>
>
> Re: Due to space limitation, the experiments for different k1 and k2 (by varying p where p=k2/k1) are moved to section A.2.5. Besides, the experiments with varying pruning rates are also shown in section A.2.5 in the supplementary.
>
>
> For the setup of other defenses, we follow the original papers to give a fair comparison, but it might be interesting to use different parameters for these works. We will consider these in the revised version.
>
>
>
>
> > Q1: What is the optimization problem that the attacker is solving?  Intuitively, my question is how does the attacker account for the ADGP defense?
>
>
>
> Re: In this paper, the attacker aims to solve the first optimization problem because the attacker wants to generate the images whose gradient is similar to the obtained gradient. Even for the second optimization problem, we can still provide theoretical proof of privacy guarantee.
> The relevant analysis is as follows.
> To simplify the expression, we use $x$ to denote the gradient and $||\cdot||$ denotes $||\cdot||_2$. [1] states the following property of topl(x) (i.e., retain the top $l$-ratio of $x$):
> $$
> ||x - {top}l(x)|| \leq \sqrt{1-l} ||x|| \tag{1}
> $$
> According to formula (1), it is easy to obtain formula (2):
> $$
> |{top}l(x)|| \ge(1-\sqrt{1-l} )||x|| \tag{2}
> $$
> Then, we make a strong assumption that ADGP(x')=ADGP(x), because the attacker has no prior knowledge of the remaining parameters, he can only randomly generate the remaining parameters, that is, E(x'-ADGP(x')))=0. So we have:
> $$
> ||x-E(x')||=||x-ADGP(x')||>||topk1(x)|||\overset{(d)}{\ge}(1-\sqrt{1-k_1} )||x||
> $$
>
>
> where (d) is directly an application of (2).
> It can be seen that our removal of the parameters of top-k1 can provide a stable lower bound on privacy.
>
>
>
> > *Q2: I also have a minor question about Definition 1. There is a nested probability and expectation. What is the source of randomness for each of these?*
>
>  Re: These sources of randomness can be divided into the randomness associated with D and randomness associated with E.
> The randomness associated with D comes from the attacker, including the choice of used optimizers (for the best optimization attack strategy), the internal randomness of the optimizers like random initialization, and etc.
> The randomness associated with E comes from a wide range of data samples, i.e., P refers to the probability distribution that the data samples need to meet for a specific learning task.
>
>
>
> References:
>
> [1] Alistarh, Dan, et al. "The convergence of sparsified gradient methods." *Advances in Neural Information Processing Systems* 31 (2018).

---

> > ### Comment · Reviewer_cSmE · 2022-08-03
> > **Thanks for your response & some further comments about the attack evaluation**
> >
> > Thanks for your response and for clarifying how the attack evaluation is performed.
> >
> > Given that the attacker constructs $x'$ to minimize $\\| \nabla W(x' ) - \mathsf{ADGP}(\nabla W(x)) \\|$, I'm concerned that this means the attacker is trying to construct $x'$ that does not correspond to any realistic data, in particular it won't look like $x$. For example, as discussed in the paper (line 142), the final layer with softmax generates large gradient coordinates. Thus ADGP will zero out these coordinates and hence the attack must produce $x'$ such that the label is unclear. It seems that the attacker is not accounting for ADGP and this results in it constructing weird $x'$.
> >
> > I'm still somewhat confused by the theoretical claims. Here's a simple (albeit silly) example that illustrates what I find odd.
> >
> > Suppose the loss is $\ell(w,x) = \left( x - \sum_i w_i \right)^2$, where $x$ is a scalar and $w$ is a vector. Then $\nabla_w \ell(w,x) = -2\left( x - \sum_i w_i \right) \cdot \vec{1}$, where $\vec{1} = (1,1,...,1)$ is a vector of $1$s the same length as $w$.
> > In this case, all of the coordinates of the gradient are the same. (If we want to make them not equal, we could set $\ell(w,x) = \left( x - \sum_i \frac{w_i}{i} \right)^2$.) So gradient pruning will zero out some arbitrary subset of the coordinates.
> >
> > No possible reconstructed $x'$ can generate a gradient that matches the pruned gradient. So in a sense this foils the attack.
> >
> > But, given *any* single coordinate of the gradient (and $\sum_i w_i$), I can reconstruct $x$, i.e., for any coordinate $j$ we have $x = \sum_i w_i - \frac12 (\nabla_w \ell(w,x) )_j$.

---

> > > ### Author Response · Authors · 2022-08-04
> > > **Further Response to cSmE**
> > >
> > > Thanks for the timely feedback. The example you kindly pointed out is more like a simplified version of a single layer perceptron and you are suggesting that partial information of the input $x$ (instead of focusing on a complete reconstruction of $x$) could be a possibility.
> > >
> > > This brings our attention back to the analytical attack discussed in Sec. 3 of the draft. We fully agree with the reviewer that, under certain conditions, directly inverting gradients to calculate the input is possible. Indeed, this is well-analyzed in the literature [1, 2] . The conclusion drawn by [1, 2] is that, the input to the network can be reconstructed uniquely from the network's gradient when a neural network containing a fully-connected (FC) layer preceded solely by (possibly unbiased) FC layers (Proposition 3.1 in [1]). Clearly, this Proposition fits perfectly with the example used by the reviewer.
> > >
> > > However, in reality, neural networks are not composed of just FC layers but of a number of nonlinear components like ReLU and pooling.
> > > So, it is difficult, if not impossible, to construct the desired analytic formula to directly invert the gradients. For such a conclusion, [1] instead proposed the best optimization-based attack, which improves the result of [2].
> > >
> > >
> > > Intuitively, in our proposed DGP and ADGP, the pruning introduces randomness as well as loss of information when the attacker wants to construct an analytic formula to directly invert the gradients. This further complicates the analytic attack from our observation. And a proof by contradiction could be easily constructed to show that if an analytic attacker can directly invert gradients for DGP (and ADGP), then this attacker should have already successfully attacked the unpruned gradients.
> > >
> > > With these considerations, our theoretical results are all based on the most advanced optimization attacks. Simply speaking, our theoretical results reveal that most advanced optimization attacks work much worse on GDP and ADGP pruned gradients when trying to recover the data.
> > >
> > > References:
> > >
> > > [1]. Geiping, Jonas, et al. "Inverting gradients-how easy is it to break privacy in federated learning?." Advances in Neural Information Processing Systems 33 (2020): 16937-16947.
> > >
> > > [2]. Phong, Le Trieu, et al. "Privacy-preserving deep learning: Revisited and enhanced." International Conference on Applications and Techniques in Information Security. Springer, Singapore, 2017.

---

> > > > ### Comment · Reviewer_cSmE · 2022-08-04
> > > > **How does Theorem 1 apply to this example?**
> > > >
> > > > Thank you for your response. I agree that the example from my previous comment is not very realistic, as we usually cannot analytically invert the gradient. But presumably Theorem 1 from the submission still applies to this simple example.
> > > >
> > > > We have an attack (given by the analytic formula) that always succeeds to perfectly reconstruct the input from the gradient in the unperturbed case. In the perturbed case where some coordinates of the gradient are zeroed out, we could pick any nonzero coordinate to successfully reconstruct; we could even use a random coordinate.
> > > >
> > > > I'm not sure how to reconcile this example with the Theorem statement. It seems like $\varepsilon=\delta=0$. What would $\gamma_1$ and $\gamma_2$ be for this example? And does the guarantee of the theorem apply to the attack?

---

> > > > > ### Author Response · Authors · 2022-08-05
> > > > > **Further Response to cSmE: How does Theorem 1 apply to this example?**
> > > > >
> > > > > We really appreciate your timely feedback and give us the chance to further clarify our contribution. To be clear, our theoretical guarantee is only aligned with Definition 1 in the main draft, which considers the optimization attack that tries to obtain a better estimate of the true gradient by optimizing a dummy data point $x$. When the estimated gradient (generated over the dummy data point $x$) exactly equals the true gradient, the attacker (i.e., central server) is deemed to be recovering the client data $x^*$ (i.e., $x = x^*$). The intuition that our DGP and ADGP provide better security is because the gradient is now pruned, and the attacker doesn't have access to the clean true (client) gradient to launch the optimization attack as it used to be. The $\epsilon$ and $\delta$ parameters in Definition 1 should be interpreted as, governed by different randomness sources, the attacker can have a good estimation (small difference $\epsilon$) of the clean true (client) gradient will be at the odds of $1-\delta$. For fixed $\delta$, a smaller $\epsilon$ indicates that the attacker is gaining an advantage, i.e., estimating the true gradient better. Theorem 1 denies this possibility.
> > > > >
> > > > > That's said, our theoretical results do not aim to address the perceived attack as pointed out by the reviewer. But this doesn't mean that the analytic attack is a real threat to our method (because this is an even weaker attack than the optimization attack as discussed in the literature [1, 2]). This can be justified further by the following observations:
> > > > >
> > > > >
> > > > > $\bullet$ The impossibility of constructing an analytic formula for a modern neural model. To check how hard it is to build the desired formula, let's just focus on how the feed-forward works for the max-pooling layer and how the gradients pass through the max-pooling layer in BP (assume all the other linear and nonlinear problems are solved). Say the forward function is $m = \max(a, b)$. In the BP stage, a known gradient value $\nabla m$ will have to be assigned to either $\partial m / \partial a$ or $\partial m / \partial b$ (if the inputs $a$ and $b$ are both known, this assignment is trivial). For the attacker who doesn't know the original inputs $a$ and $b$, it will bring two different possibilities. The complexity grows exponentially to $4^n$  or $9^n$ (with $n$ being the number of used max-pooling operators) for typical models in real life.
> > > > >
> > > > >
> > > > > $\bullet$ Any method that can analytically solve the DGP (or ADGP) (to recover the data $x^*$) indicates a solution to the unpruned case, which is deemed as intractable in the literature for the reasons discussed above (indeed, it is not possible to list all the formulas when the size increases exponentially). Take the case of DGP as an example, and assume the attacker can recover data $x^*$ from DGP($\nabla \mathbf{W}$). Considering that DGP($\nabla \mathbf{W}$) is just a pruned version of $\nabla \mathbf{W}$ (i.e., $\nabla \mathbf{W}$ = DGP($\nabla \mathbf{W}$) + $\mathbf{e}$), the attacker can always optimize this gradient noise term $\mathbf{e}$ (with an auxiliary polynomial time optimization-based attack) and call this ideal analytically solver to recover $x^*$ from $\nabla \mathbf{W}$. In other words, if the ideal analytical solver exists for the case of DGP, then it will solve the exponential-sized problem with the help of a polynomial time optimizer, which indicates a contradiction ($P = NP$ in computational theory).
> > > > >
> > > > >
> > > > > Overall, the optimization-based attacks we considered imposed a real threat of inverting the gradients on modern neural models, and our theoretical analyses all aligned with this threat. The analytical method remains to be an open challenge (for even just constructing the formula in polynomial-time as the starting point). **We sincerely expect the response clarifies the reviewer's concern.**

---

> > > > > > ### Comment · Reviewer_cSmE · 2022-08-08
> > > > > > **Thank you, but still concerned.**
> > > > > >
> > > > > > I thank the authors for their explanations. However, I remain concerned about this work.
> > > > > >
> > > > > > I agree that the example I gave is simpler than any realistic deep learning application. However, it still highlights a flaw in the analysis presented in the paper. Even if there is no analytic formula for inverting the gradient, the same problem could arise with realistic attacks.
> > > > > >
> > > > > > It seems that the theoretical and experimental results are both showing that there is no $x'$ such that $\\|\nabla W(x') - \mathsf{ADGP}(\nabla W(x))\\|$ is small. However, this does not imply that gradient inversion attacks are foiled by ADGP. The issue is that $\\|\nabla W(x') - \mathsf{ADGP}(\nabla W(x))\\|$ being large does not imply $\\|x'-x\\|$ is large. Thus the guarantees shown in the paper do not imply effective defense against gradient inversion attacks.
> > > > > >
> > > > > > The general practice in security research when defense schemes like this are proposed is to be skeptical, as the historical success rate is low.
> > > > > >
> > > > > > I am skeptical of this defense. As a general rule of thumb, inversion-style attacks work when the dimension of the released output is higher than the dimension of the data to be reconstructed. With overparameterized neural networks, the gradient pruning proposed here will not sufficiently reduce the output dimension. As the authors point out, there is potentially a computational barrier to inversion, but research has shown that gradient inversion is often possible, and I see no reason to believe that ADGP makes this task more computationally infeasible than without any defense.

---

> > > > > > > ### Author Response · Authors · 2022-08-09
> > > > > > > **Further Response to Reviewer cSmE: Further explanation**
> > > > > > >
> > > > > > > Thanks for your response. Here is the further explanation.
> > > > > > > We cannot provide a specific theoretical analysis based on this analytical attack because, as we discussed in the previous response, it is impossible to list the exact attack equations. But the attack you are concerned about does exist [22, 40], and we have covered this attack in Section 3 and used the state-of-the-art work [22] to evaluate the defenses in experiments. The recovered data results are shown in Fig.1(a), our method can provide effective privacy protection by preventing the attacker from recovering data. And there could be a misunderstanding in the following sentence.
> > > > > > >
> > > > > > > > It seems that the theoretical and experimental results are both showing that there is no $x'$ such that $\|\nabla W(x') - \mathsf{ADGP}(\nabla W(x))\|$ is small. However, this does not imply that gradient inversion attacks are foiled by ADGP.
> > > > > > >
> > > > > > > Although Definition 1 and Theorem 1 only involve the distance between gradients and not the recovered data, Proposition 1 shows that the quality of the data that the attacker can recover is directly related to the distance between gradients, i.e., ADGP can reduce the quality of recovered data by increasing the distance between gradients.
> > > > > > >
> > > > > > > Furthermore, in Section 6.2, we evaluate privacy by comparing the quality of data recovery rather than the distance between gradients, which demonstrates our effective defense.
> > > > > > >
> > > > > > > But this is indeed an oversight, as we did not indicate this in Definition 1 and Theorem 1. We will explain this in the revised version.
> > > > > > >  **We sincerely expect the response clarifies the reviewer's concern.**

---

### Meta-Review · Area_Chair_xkDh · 2022-08-20

**Recommendation:** Reject
**Confidence:** Less certain

**Metareview:**

This paper proposes a method for defending against gradient inversion attacks.
A gradient inversion attack attempts to reconstruct the training data from the model and its gradient.
Gradient inversion attacks have been performed in practice, which demonstrates that sharing gradients rather than raw data provides limited privacy protection.

The proposed method,"Aligned Dual Gradient Pruning (ADGP)," perturbs the gradients by zeroing out a large subset of the coordinates including both small and large values.

The key claim is that this prevents reconstruction of the training data.
The paper provides both theoretical and experimental results to support this claim.

However, these results rely on implicit assumptions about the form of the gradient inversion attack. Specifically, they assume a vanilla gradient inversion attack that does not compensate for the ADGP defense. In particular, ADGP creates sparse gradients and it is assumed that the attacker attempts to reconstruct an input whose unperturbed gradient is sparse, even if the true unperturbed gradient is not sparse.

This assumption is a form of "security through obscurity." We should assume that the attacker is aware of the defense and tailors the reconstruction to the defense. Thus theoretical/empirical evaluation should consider attacks that are designed specifically for ADGP.

Overall, the key claim of the paper is not adequately supported (and, in my opinion, it seems likely that the proposed defense is not effective). Thus this paper should not be published.

**Award:**

No

---

### Decision · Program_Chairs · 2022-09-14

Reject